# Unusual interlayer quantum transport behavior caused by the zeroth Landau level in YbMnBi$_2$

J.Y. Liu[1], J. Hu [1], D. Graf[2], T. Zou[3], M. Zhu[3], Y. Shi[4], S. Che[5], S.M.A. Radmanesh[6], C.N. Lau[5], L. Spinu[6], H.B. Cao[7], X. Ke[3] & Z.Q. Mao[1]

Relativistic fermions in topological quantum materials are characterized by linear energy–momentum dispersion near band crossing points. Under magnetic fields, relativistic fermions acquire Berry phase of $\pi$ in cyclotron motion, leading to a zeroth Landau level (LL) at the crossing point, a signature unique to relativistic fermions. Here we report the unusual interlayer quantum transport behavior resulting from the zeroth LL mode observed in the time reversal symmetry breaking type II Weyl semimetal YbMnBi$_2$. The interlayer magnetoresistivity and Hall conductivity of this material are found to exhibit surprising angular dependences under high fields, which can be well fitted by a model, which considers the interlayer quantum tunneling transport of the zeroth LL's Weyl fermions. Our results shed light on the unusual role of zeroth LLl mode in transport.

[1] Department of Physics and Engineering Physics, Tulane University, New Orleans, LA 70118, USA. [2] National High Magnetic Field Lab, Tallahassee, FL 32310, USA. [3] Department of Physics and Astronomy, Michigan State University, East Lansing, MI 48824, USA. [4] Department of Physics, University of California, Riverside, CA 92521, USA. [5] Department of Physics, The Ohio State University, 191 West Woodruff Avenue, Columbus, OH 43210, USA. [6] Department of Physics and Advanced Materials Research Institute, University of New Orleans, New Orleans, LA 70148, USA. [7] Quantum Condensed Matter Division, Oak Ridge National Laboratory, Oak Ridge, TN 37831, USA. Correspondence and requests for materials should be addressed to J.H. (email: jhu@tulane.edu) or to Z.Q.M. (email: zmao@tulane.edu)

n conventional metals, the energy of the quantized Landau level (LL) increases linearly with increasing magnetic field. However, in topological materials such as graphene[1, 2] and recently discovered Dirac/Weyl semimetals $Cd_3As_2$[3–7], $Na_3Bi$[8, 9] $ZrTe_5$,[10, 11] and TaAs-type monopnictides[12–18], the quantized energies of LLs are given by $\varepsilon_n = \pm v_F\sqrt{2e\hbar B|n|}$ ($n = 0, \pm 1, \pm 2\ldots$) for two-dimensional (2D) Dirac/Weyl fermions[19] or $\varepsilon_n = \pm v_F\sqrt{2e\hbar B|n| + k_z^2}$ ($n = 0, \pm 1, \pm 2\ldots$, $k_z$ is the momentum along the field direction) for three-dimensional (3D) cases[20]. The $n = 0$ level corresponds to the zeroth energy LL, which is a signature unique to topological fermions but absent in non-relativistic electron systems. For 2D Dirac/Weyl fermions, the zeroth LL is always locked to the band crossing point (i.e., the Dirac/Weyl node) upon field sweep. However, for 3D cases, the energy of the zeroth LL disperses linearly with $k_z$. For a given topological material, if the Dirac/Weyl node is away from the Fermi energy $E_F$, its $n \neq 0$ LLs would successively pass through $E_F$ upon increasing magnetic field, thus resulting in oscillating density of state (DOS) at $E_F$, which can be probed through quantum oscillations in resistivity or magnetic susceptibility. In this case, the zeroth LL manifests itself in the phase shift (i.e., Berry phase) in quantum oscillations[21, 22]. When the quantum limit is approached, the zeroth LL could lead to new exotic phenomena, e.g., dynamic mass generation in $ZrTe_5$[23]. By contrast, if the Dirac/Weyl node is at $E_F$, no $n \neq 0$ LLs pass $E_F$ upon increasing the field, and the $DOS(E_F)$ is contributed only by the zeroth LL. Under this circumstance, the $DOS(E_F)$ would monotonically increase due to the increase of the zeroth LL's degeneracy. In general, it is hard to observe such an effect in transport measurements in most topological materials due to their Dirac/Weyl nodes away from $E_F$ and/or the complexity of multiband electronic structure. In this paper, we report the unusual quantum transport behavior directly arising from the zeroth LL in the time reversal symmetry (TRS) breaking Weyl semimetal $YbMnBi_2$[24]: the zeroth LLs' Weyl fermions contribute to interlayer transport through quantum tunneling.

$YbMnBi_2$ shares a similar layered structure with $SrMnBi_2$ and $EuMnBi_2$, which have been established as Dirac materials[25, 26] with interesting properties (e.g., the valley-polarized interlayer conduction in $SrMnBi_2$[27] and the quantum Hall effect due to the magnetically confined 2D Dirac fermions in $EuMnBi_2$[26]). One common character of these materials is that their Weyl/Dirac fermions are generated by the 2D Bi square-net planes. The Weyl state in $YbMnBi_2$ is believed to originate from the TRS breaking caused by a ferromagnetic component of the canted anti-ferromagnetic order developed by the Mn sublattice[24]. The electronic band structure of $YbMnBi_2$ is of a quasi-2D character due to its layered crystal structure. Angle-resolved photoemission spectroscopy (ARPES) has revealed that the Fermi surface of this material consists of the hole-like and electron-like pockets comprised of linear Dirac bands. At the connection points of electron- and hole-like pockets, type II Weyl points with the nodes at $E_F$ have been observed[24].

In our work, we take advantage of the quasi-2D electronic structure of $YbMnBi_2$ as well as its Weyl nodes at $E_F$ to probe the transport properties of Weyl fermions at the zeroth LLs. Considering the 2D Landau quantization in $YbMnBi_2$, the presence of Weyl nodes at $E_F$ not only leads the zeroth LLs of the Weyl bands to appear at $E_F$ so that the Weyl fermions at the zeroth LLs directly participate in transport but also makes the quantum limit of Weyl bands accessible at a relatively low field, which is important to observe the transport properties of the zeroth LLs' Weyl fermions in a multiple band system (note that when a Weyl node is at $E_F$, the quantum limit of the Weyl bands can be

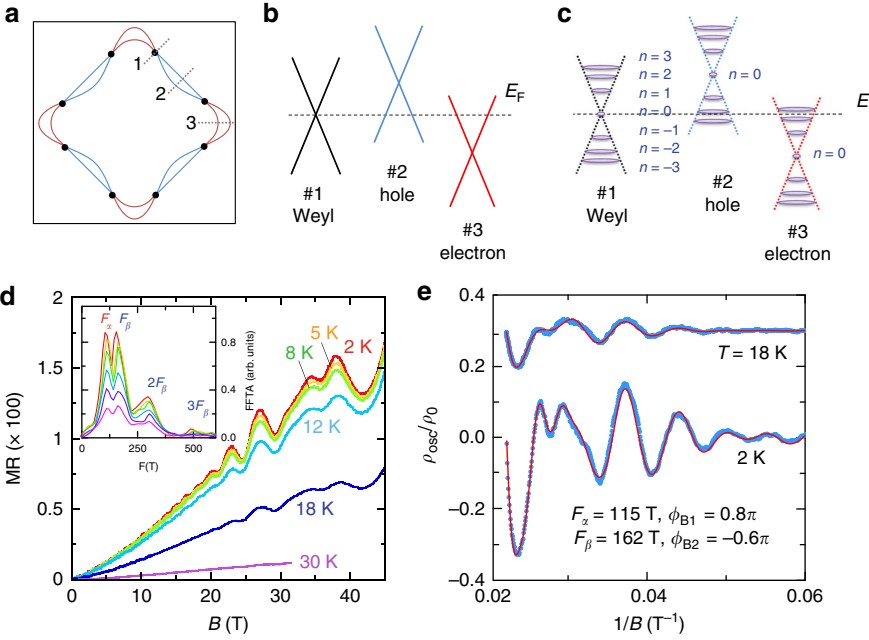

**Fig. 1** Schematic electronic band structure and in-plane magnetotransport properties of $YbMnBi_2$. **a** Schematic of $YbMnBi_2$'s Fermi surface determined by ARPES experiments[24]. The *red* and *blue pockets* correspond to electron- and hole-like pockets, respectively. The *black dots* represent Weyl points. **b** Schematic of the linear band crossing for the electron- and hole-like pockets and the Weyl point, also determined by ARPES experiments for the cuts 1–3 shown in **a**[24]. **c** Schematic of Landau levels for three types of band crossings shown in **b** under high magnetic fields. We adopted the 2D Landau quantization mode because of the quasi-2D electronic structure of $YbMnBi_2$. **d**, The normalized in-plane magnetoresistivity MR [= $\frac{\rho_{xx}(\mathbf{B}) - \rho_{xx}(\mathbf{B}=0)}{\rho_{xx}(\mathbf{B}=0)}$] as a function of magnetic field along the out-of-plane direction. *Inset*, the FFT spectra of the SdH oscillations. **e** The fits of SdH oscillations at 2 and 18 K by the two-band LK formula (see the Methods section for more details for the fits). The SdH oscillatory component $\rho_{osc}$ is obtained by subtracting the magnetoresistivity background. $\rho_0$ is the zero-field resistivity

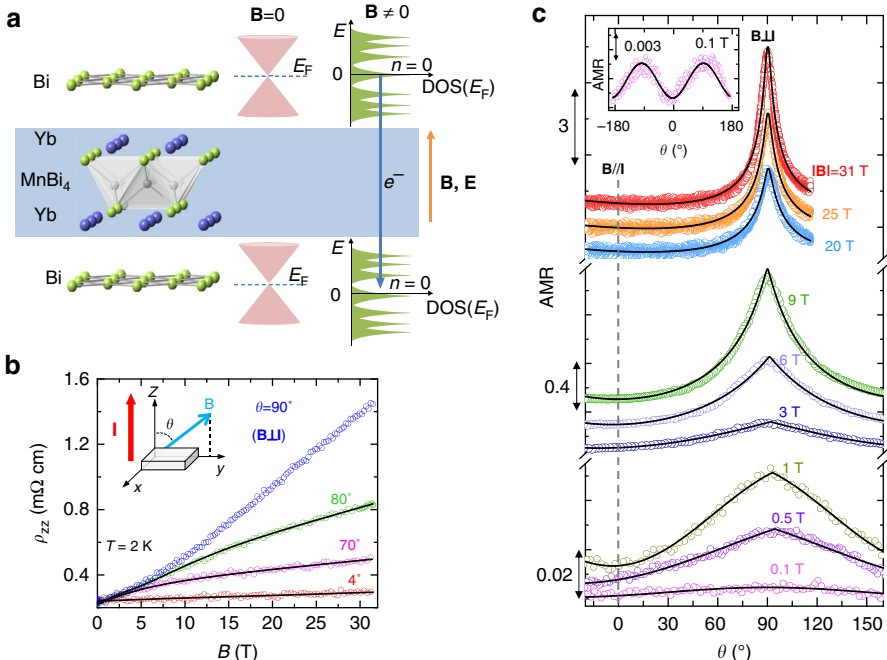

**Fig. 2** Interlayer magnetotransport properties of YbMnBi$_2$. **a** Schematic of the interlayer tunneling of the zeroth LLs' Weyl fermions. **b** The field dependence of the out-of-plane resistivity, $\rho_{zz}(\mathbf{B})$, under different field orientations at $T = 2$ K. The *inset* shows the experimental setup. The *solid lines* superimposed on the data represent the fits to Eq. (2) in the text. The fit for $\theta = 90°$ is not available since the zeroth LLs disappear for in-plane field. **c** Angular dependence of magnetoresistance (AMR), measured under different fields up to 31 T and at $T = 2$ K. The *black curves* superimposed on the data represent the fits to Eq. (2) in the text. At low fields (e.g., 0.1 T), AMR shows the $\sin^2\theta$ dependence expected for the Lorentz effect as shown in the *inset*, indicating that the interlayer transport at low fields is dominated by the Dirac band transport as discussed in the text

reached as long as the zeroth LL is distinguishable from other LLs). The quasi-2D electronic structure provides us with an opportunity to tune the DOS($E_F$) contributed by the Weyl points via controlling the zeroth LL's degeneracy of the Weyl bands by rotating the magnetic field from the out-of-plane to in-plane direction. In our experiments, we measured the angular dependences of various longitudinal and Hall resistivities to reveal the role of the zeroth LL in transport. We observe very unusual behaviors in these experiments, which can be well fitted by a model that considers both the interlayer quantum tunneling transport of the zeroth LLs' Weyl fermions and the momentum relaxation transport of the Dirac fermions hosted by hole- and electron-like pockets.

## Results

**Material characterization and in-plane transport measurements.** The YbMnBi$_2$ single crystals were synthesized using a flux method (see Methods). We have performed neutron-scattering experiments on YbMnBi$_2$ single crystals, which not only confirmed its tetragonal lattice structure (see Supplementary Table 1 for detailed structural parameters) but also revealed a C-type antiferromagnetic state below $T_N = 298$ K, with the ordered moment of 3.789(3) $\mu_B$ per Mn (Supplementary Fig. 1), in agreement with the magnetic structure reported previously by Wang et al.[28]. The Yb spins do not order even down to 4 K. Although we also observed very weak ferromagnetism in the magnetization measurements (Supplementary Fig. 2), consistent with the report by Borisenko et al.[24], it could not be resolved in neutron-scattering experiments within the instrumental resolution. As also seen by Wang et al.[28], the measured in-plane ($\rho_{xx}$) and out-of-plane ($\rho_{zz}$) resistivity reveal anisotropic electronic properties (Supplementary Fig. 3). Although both $\rho_{xx}$ and $\rho_{zz}$ exhibit metallic temperature dependences, their anisotropic ratio

$\rho_{zz}/\rho_{xx}$ reaches 36 at $T = 2$ K, suggestive of a moderately anisotropic electronic structure.

To better interpret our transport data presented below, we first show the schematics of the Fermi surfaces projected on the $k_x$–$k_y$ plane of YbMnBi$_2$ and its band dispersions along several typical momentum directions determined by the previous ARPES experiments[24], respectively, in Fig. 1a, b. The Fermi surface of YbMnBi$_2$ consists of the Weyl points at $E_F$ (denoted by *black dots* in Fig. 1a) and the hole-like (marked in *blue*) and electron-like (*red*) pockets comprised of linear Dirac bands. The Weyl points appear at the momentum points where electron- and hole-like pockets are connected, which is a typical signature of type II Weyl semimetal[29]. Given such multiband electronic structure, the transport properties of YbMnBi$_2$ should be contributed by both the Weyl points and the hole- and electron-like pockets. Due to the fact that the Weyl nodes are at $E_F$, while the Dirac bands forming the hole- and electron-like pockets cross at energies above or below $E_F$ as illustrated in Fig. 1b, the Weyl and Dirac bands are expected to exhibit distinct magnetotransport behaviors according to the above discussions. For the Dirac bands, since their crossing points, i.e., the Dirac nodes, are away from $E_F$ (Fig. 1b), their zeroth LLs are at the Dirac nodes rather than at $E_F$ (Fig. 1c). In this case, quantum oscillations are expected upon increasing magnetic field. However, for the Weyl bands with their crossing points at $E_F$ (Fig. 1b), since the 2D LL quantization leads the zeroth LLs to be pinned to $E_F$ regardless of magnetic field strength (Fig. 1c), increasing magnetic field along any direction is not expected to result in quantum oscillations, but leads to a monotonic increase in the DOS($E_F$) of the Weyl bands if the field is not within the plane. In our magnetotransport measurements, we indeed observed signatures expected for both the Dirac and Weyl bands, as will be shown below.

Figure 1d shows the normalized in-plane magnetoresistivity MR, defined as $\frac{\rho_{xx}(\mathbf{B}) - \rho_{xx}(\mathbf{B}=0)}{\rho_{xx}(\mathbf{B}=0)}$, as a function of magnetic field (up to

45 T) measured at various temperatures for YbMnBi$_2$. Remarkable Shubnikov-de Haas (SdH) oscillations can be seen from these data at low temperatures. The fast Fourier transform (FFT) analyses for the oscillatory components $\rho_{osc}$ reveal two oscillation frequencies, i.e., $F_\alpha = 115$ T and $F_\beta = 162$ T (see the *inset* to Fig. 1d). We note Wang et al.[28] previously reported the SdH oscillations of $\rho_{xx}$ for YbMnBi$_2$, but the FFT spectrum derived from their data shows only a broad peak at about 130 T, contrasted with our observation of two frequencies at 115 and 162 T. Such an inconsistency may be due to the fact that their magnetorsistivity measurements were made only up to 35 T; the limited field range makes it hard to precisely resolve oscillation frequencies. The SdH oscillation patterns in our data (Fig. 1d) show remarkable features resulting from multiple oscillation frequencies above 30 T, i.e., the oscillation peaks are not equally spaced on the scale of $1/B$ as shown in Fig. 1e, clearly indicating that the double frequencies ($F_\alpha$ and $F_\beta$) revealed in our FFT spectra (*inset* to Fig. 1d) are intrinsic. From the analyses of SdH oscillations, we derived Dirac fermion properties. From the fits of temperature dependences of the FFT amplitude by the thermal damping factor of the Lifshitz–Kosevich (LK) formula[30, 31], i.e., $\frac{2\pi^2 k_B T m^*/\hbar e|\mathbf{B}|}{\sinh(2\pi^2 k_B T m^*/\hbar e|\mathbf{B}|)}$ (see Methods and Supplementary Fig. 4), the effective cyclotron masses $m^*$ associated with the oscillation frequencies $F_\alpha$ and $F_\beta$ are estimated to be $\sim 0.24m_0$ ($m_0$, the free electron mass). As noted above, the Berry phase of $\pi$ accumulated in cyclotron motion is the fundamental topological property of relativistic fermions. However, for YbMnBi$_2$, it is hard to precisely determine the Berry phase using the commonly accepted LL fan diagram due to the existence of multiple oscillation frequencies in SdH oscillations. We evaluated the Berry phases of YbMnBi$_2$ through the direct fits of the oscillation patterns by the multiband LK formula[32] (see Methods). As shown in Fig. 1e, the SdH oscillation patterns at 2 and 18 K can be best fitted by the two-band LK model when the higher harmonic components $2F_\beta$, $3F_\beta$ and $4F_\beta$ revealed in FFT were included in the fits (note that the $4F_\beta$ component is weak and not shown in the *inset* to Fig. 1d). The extracted Berry phases from these fits are $0.8\pi$ for the $F_\alpha$ bands and $-0.6\pi$ for the $F_\beta$ bands. This result is based on the assumption that both $F_\alpha$ and $F_\beta$ bands are exactly 2D. Given that the quasi-2D electronic band structure of YbMnBi$_2$, an additional phase factor of $\pm 0.25\pi$ should be taken into account[31]; thus, the Berry phase would be $0.8\pi \pm 0.25\pi$ for $F_\alpha$ bands and $-0.6\pi \pm 0.25\pi$ for $F_\beta$ bands. In either case, the fitted Berry phases are clearly nontrivial.

**Interlayer transport measurements.** From the electronic band structure of YbMnBi$_2$ introduced above (Fig. 1a–c), it is apparent that the SdH oscillations observed in $\rho_{xx}(\mathbf{B})$ result from the Dirac bands. Our above demonstration of nontrivial Berry phases provides clear transport evidence for the Dirac fermions hosted by these bands. As discussed above, for 2D LL quantization, the Weyl points at $E_F$ shown in Fig. 1a would not give rise to any quantum oscillations. Since the zeroth LLs of Weyl cones are pinned at $E_F$ (Fig. 1c), their increased degeneracy upon increasing magnetic field would cause DOS($E_F$) to increase monotonically as noted above. This effect, though hardly causing any noticeable features in the in-plane magnetoresistance, results in peculiar signatures in the field orientation dependence of interlayer magnetotransport, as we will show below.

In Fig. 2b, c, we, respectively, present the field dependences of interlayer magnetoresistivity $\rho_{zz}(\theta,\mathbf{B})$ under various field orientations and the angular dependences of interlayer magnetoresistivity AMR [defined as $\frac{\rho_{zz}(\theta,\mathbf{B})-\rho_{zz}(\theta=0,\mathbf{B})}{\rho_{zz}(\theta=0,\mathbf{B})}$] under different fields at 2 K. The *inset* to Fig. 2b illustrates our experimental setup. Both the $\rho_{zz}(\mathbf{B})$ and AMR data exhibit anomalous features attributable to

the quantum transport of the zeroth LL's Weyl fermions. First, $\rho_{zz}(\mathbf{B})$ displays sublinear field dependence as the field is tilted toward the z-axis ($\theta < 90°$), in contrast with the scenario of $\theta = 90°$, where $\rho_{zz}(\mathbf{B})$ exhibits $B^2$ dependence in a low-field region, but gradually evolves to a linear field dependence above 10 T (Fig. 2b). Such an unusual evolution of $\rho_{zz}(\mathbf{B})$ with $\theta$ cannot be understood in light of the classical orbital effect or other quantum effects such as weak anti-localization as discussed in Supplementary Note 1. Given that 2D LL quantization is absent for $\theta = 90°$ but gradually develops with decreasing $\theta$ for $\theta < 90°$, the unusual sublinear field dependence of $\rho_{zz}(\mathbf{B})$ seen at $\theta < 90°$ is associated with LL quantization as discussed below. Second, AMR (Fig. 2c) exhibits very unusual angular dependence under high fields. At a field of 31 T, we observed a very sharp peak at $\theta = 90°$ ($\mathbf{B} \perp \mathbf{I}$) and nearly angle-independent magnetoresistivity below $\theta = 60°$. With decreasing the field, the peak becomes gradually suppressed and broadened; significant suppression and broadening are observed below 9 T. Surprisingly, when $|\mathbf{B}| < 1$ T, AMR evolves into $\sin^2\theta$ dependence as shown by the *solid fitted curves* (e.g., see the data taken at 0.1 T in the *inset* to Fig. 2c), in contrast with AMR at high fields. The $\sin^2\theta$ dependence of magnetoresistivity is generally expected for the classical orbital effect for which AMR $(\theta) \propto B_{xy}^2 = \mathbf{B}^2\sin^2\theta$. The strong deviation of AMR from the $\sin^2\theta$ dependence in the high-field range implies that the interlayer transport mechanism in the high-field range is distinct from that in the low-field range.

**Discussions**

Next we will show it is the zeroth LLs of the Weyl bands that make the interlayer transport under high fields distinct from the low-field interlayer transport. As indicated above, the Fermi surface of YbMnBi$_2$ consists of not only the Weyl points at $E_F$ but also the hole- and electron-like pockets comprised of linear Dirac bands. Therefore, both the Dirac and Weyl bands should contribute to the interlayer magnetotransport in YbMnBi$_2$. We will first consider the contribution from the Weyl bands. Since the Weyl bands' zeroth LL is locked to $E_F$ (Fig. 1c), the Weyl bands' contribution to transport should come only from the Weyl fermions at the zeroth LLs. Given that the first-principle calculations predicted the electronic states near $E_F$ are all contributed by the 2D Bi square-net layers and the Weyl bands are of 2D character[24], we can reasonably assume the interlayer transport of the zeroth LLs' Weyl fermions in YbMnBi$_2$ takes place through quantum tunneling process as depicted in Fig. 2a. In this case, the tunneling current of the zeroth LLs' Weyl fermions is highly sensitive to the magnetic field and its orientation. If the field is oriented along the out-of-plane direction and sweeps to a large magnitude, the tunneling current would enhance remarkably, owing to the increase of DOS($E_F$) induced by the enhanced zeroth LLs' degeneracy. Since the quasi-particle's cyclotron motion is confined within the plane in a 2D limit, rotating the field away from the out-of-plane direction would suppress LL quantization, which reduces the zeroth LLs' degeneracy, thus resulting in the decrease of tunneling conductivity. Such a phenomenon has been demonstrated in the pressurized layered organic conductor $\alpha$-(BEDT-TTF)$_2$I$_3$, which has a 2D Dirac cone with the node being exactly at $E_F$ in each BEDT-TTF molecular layer[33, 34]. According to ref. [33], the tunneling conductance $\sigma_t^{LL0}$ due to the zeroth LLs in a multilayer relativistic fermion system can be described by

$$\sigma_t^{LL0} = A \cdot |B\cos\theta| \exp\left[-\frac{1}{2}\frac{ed^2(B\sin\theta)^2}{\hbar|B\cos\theta|}\right] \quad (1)$$

where $A$ is a field-independent parameter and $d$ is the interlayer spacing of the neighboring layers hosting relativistic fermions. When we apply this tunneling model to YbMnBi$_2$, $d$ should be

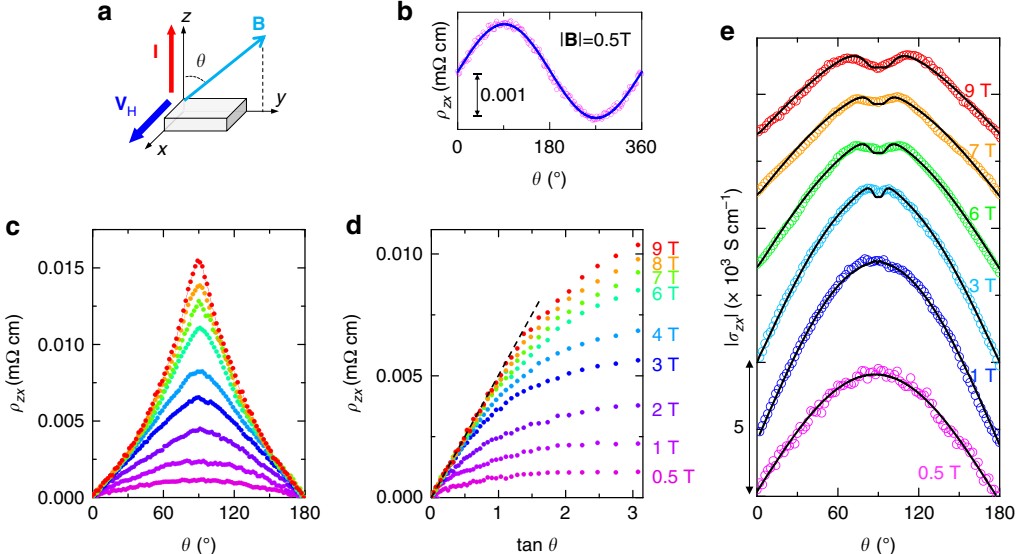

**Fig. 3** Interlayer Hall effect for YbMnBi$_2$. **a** Experimental setup for Hall effect measurements. **b** Angular dependence of Hall resistivity $\rho_{zx}$ at $B = 0.5$ T and $T = 2$ K, which follows a $\sin\theta$ dependence as indicated by the *fitted solid curve*. **c** Angular dependence of $\rho_{zx}$ at various fields from 0.5 to 9 T. **d** $\rho_{zx}$ plotted against $\tan\theta$; the $\tan\theta$ asymptote (i.e., the *dashed line*) can be observed at low angles. **e** Angular dependence of Hall conductivity $\sigma_{zx}$ at $T = 2$ K under different magnetic fields (the conversion process from the measured resistivity tensor elements to $\sigma_{zx}$ is shown in Methods). The data at different fields have been shifted for clarity. The *solid lines* represent the fits by Eq. (3)

the spacing between the neighboring 2D Bi square-net planes (Fig. 2a), which is equal to 1.0824 nm according to our neutron-scattering measurements (Supplementary Table 1). As shown below, such a tunneling model based on the zeroth LLs provides an excellent interpretation for the unusual interlayer magneto-transport behavior described above for YbMnBi$_2$.

To make quantitative fits to the $\rho_{zz}(\mathbf{B})$ and AMR($\theta$) data in Fig. 2b, c using the above tunneling model, we have to take the Dirac bands' contribution to the interlayer transport into account. As discussed above, the SdH oscillations probed in the in-plane magnetoresistivity (Fig. 1d) indeed reflect the Dirac bands' contribution to the in-plane transport. Similar SdH oscillations due to the Dirac bands are also expected in the interlayer mag-netoresistivity. However, this is not observed experimentally, as shown in Fig. 2b. This can probably be attributed to anisotropic mobility of Dirac fermions in YbMnBi$_2$. We assume the Dirac bands contribute to the interlayer transport through a momen-tum relaxation mechanism (i.e., coherent band transport). This assumption is based on the fact that YbMnBi$_2$ exhibits a moderate electronic anisotropy as reflected in the $\rho_{zz}/\rho_{xx}$ resistivity ratio (∼ 36 at $T = 2$ K; Supplementary Fig. 3). If the Dirac bands were also highly 2D-like as the Weyl bands are, a large electronic aniso-tropy would be expected, inconsistent with the experimental observation. When we combine the Dirac fermion transport through momentum relaxation with the quantum tunneling transport of the zeroth LLs' Weyl fermions (Fig. 2a), the overall interlayer magnetoresistivity under a field oriented at an angle $\theta$ can be expressed as

$$\rho_{zz}(\mathbf{B}, \theta) \approx 1/\sigma_{zz}(\mathbf{B}, \theta) = 1/\left[\sigma_t^{LL0}(\mathbf{B}, \theta) + \sigma_c(\mathbf{B}, \theta)\right] \quad (2)$$

where $\sigma_t^{LL0}$ represents the tunneling conductivity of the zeroth LLs given by Eq. (1) and $\sigma_c$ stands for the conductivity due to the Dirac bands' transport. In general, $\sigma_{zz}$ should be obtained via taking the inverse of the resistivity tensor (see Methods). Our resistivity–conductivity tensor conversion analyses demonstrate the assumption of $\sigma_{zz} \approx 1/\rho_{zz}$ in Eq. (2) is valid for our experi-mental setup, as shown in Supplementary Fig. 6c. $\sigma_c$ in Eq. (2) can be derived from the field dependence of magnetoresistivity at $\theta =$

90°. As seen in our experiment setup (see the *inset* to Fig. 2b), at $\theta = 90°$, the tunneling transport of the zeroth LLs should vanish due to the absence of quantized LLs, so that the interlayer transport should be mostly dominated by the momentum relaxation of the Dirac bands. As shown in Fig. 2b, the $\rho_{zz}(\mathbf{B}, \theta = 90°)$ exhibits a quadratic field dependence in a low-field range, but crossover to a linear field dependence at high fields. If we assume interband scattering is negligible, the inter-layer magnetoresistivity of the Dirac band transport channel can be assumed to follow the same trend even when the tunneling transport channel of the zeroth LLs of the Weyl bands sets in for $\theta < 90°$. Thus, in Eq. (2), $\sigma_c$ can be taken as $\sigma_0/(1 + k_1 \cdot B_{xy}^2)$ ($\sigma_0$, the Drude conductivity; $k_1$, a constant; $B_{xy} = |\mathbf{B}|\sin\theta$) for low fields, but as $\sigma_0/(1 + k_2 \cdot |B_{xy}|)$ for high fields. The validity of this treatment is demonstrated in Supplementary Fig. 6d. With these approximations, we can reproduce all the field- and angle-dependent interlayer magnetoresistivity data shown in Fig. 2b, c using Eq. (2). The *solid lines* in Fig. 2b, c represent our fitting curves.

Intuitively, one may expect negative longitudinal magnetore-sistance (LMR) for $\mathbf{B}//\mathbf{I}$ (i.e., $\theta = 0°$), since the tunneling transport channel should give rise to negative magnetoresistance as reflected in Eq. (1) and the classical orbit magnetoresistance due to Lorentz effect is absent for $\mathbf{B}//\mathbf{I}$. However, we observed a weak positive magnetoresistance for $\theta = 4°$ (Fig. 2b). Such a result can be understood in term of the competition of the positive mag-netoresistance component of the momentum relaxation channel and the negative component of the tunneling channel. In general, large positive LMR is a generic feature of topological semimetals. For example, AMn(Bi/Sb)$_2$ (A = Sr, Ba, Ca), which are iso-structural to YbMnBi$_2$ and Dirac materials, have remarkable positive LMR for interlayer transport. LMR reaches a few hun-dreds percent for BaMnBi$_2$[35], BaMnSb$_2$[36], and SrMnSb$_2$[37] at 9 T and ∼ 2 K, even as high as 10,000% at 31 T[37]. Such large positive LMR can be attributed to their Dirac band transport. Since the Dirac nodes in these materials are far away from the Fermi level, their interlayer transport should not involve the zeroth LL's tunneling. Therefore, we can reasonably expect a very large positive LMR component resulting from the Dirac band transport

channel in YbMnBi$_2$ due to its structural similarity to AMn(Bi/Sb)$_2$. However, our observed LMR for $\rho_{zz}$ in YbMnBi$_2$ reaches only 20% even at 31 T (Fig. 2b), which are several orders of magnitude smaller than those of AMn(Bi/Sb)$_2$ materials at the same field[35–37]. The strong suppression of positive LMR in YbMnBi$_2$ implies that its large positive LMR component expected for the Dirac band transport channel must be canceled by a large negative magnetoresistance component caused by the zeroth LL tunneling of the Weyl bands.

The evolution of AMR from the $\sin^2\theta$ dependence at low fields to the sharp peak at $\theta = 90°$ above 9 T (Fig. 2c) can now be well understood in light of the theoretical fits based on Eq. (2). At low fields, the electron- and hole-like Fermi pockets (Fig. 1a) should make dominant contributions to the transport, since these pockets should have a much greater DOS($E_F$) than the Weyl points. The observation of the $\sin^2\theta$ dependence of AMR at low fields implies that the contribution of the hole- and electron-like pockets to AMR follows the classic Lorentz effect for which the interlayer magnetoresistivity is proportional to $B_{xy}^2$ [= (|**B**|$\sin\theta$)$^2$]. When the field is remarkably increased, the DOS($E_F$) of the Weyl points should increase dramatically. This is because that the Weyl nodes are at $E_F$ in YbMnBi$_2$ as indicated above, such that quantum limit of Weyl bands should be reached under a relatively low magnetic field, when the energy spacing between the zeroth and first LL is greater than the LL's breadth. Near the quantum limit of the Weyl bands, the zeroth LLs' degeneracy would enhance significantly, thus resulting in significantly increased DOS($E_F$) at the zeroth LLs and enhanced tunneling conductivity. The gradual deviation from the $\sin^2\theta$ dependence in AMR upon increasing field suggests that the Weyl fermions at the zeroth LLs play a more important role in interlayer transport under high fields. Our successful fits of the $\rho_{zz}(\mathbf{B})$ and AMR($\theta$) data to Eq. (2) strongly support that the zeroth LL's Weyl fermions contribute to the interlayer transport via a tunneling process.

Although the LL degeneracy of Dirac bands is also enhanced upon increasing field, it should not contribute to the unusual features of AMR at high fields shown in Fig. 2c. Since the quantum oscillation frequencies of Dirac fermions are high (115 and 162 T; Fig. 1d), the quantum limit of the Dirac bands cannot be reached until the field is increased above 230 T. Given that our experiments were conducted below 31 T, the variation of LL degeneracy should be small for the Dirac bands. Therefore, the variation of the DOS($E_F$) of the Dirac bands with the field rotation in the field range of our experiments is expected to be small and the AMR of the Dirac fermion transport channel should more or less follow the classical Lorentz effect, i.e., AMR($\theta$) $\propto B_{xy}^2 = \mathbf{B}^2\sin^2\theta$, which is only observed at low fields as indicated above.

Our argument of the tunneling transport of the zeroth LLs' Weyl fermions is further corroborated by the measurements of the dependence of Hall resistivity $\rho_{zx}$ on field orientation for YbMnBi$_2$. We note such an experimental approach was used to demonstrate the interlayer tunneling of the zeroth LL' Dirac fermions in the pressurized layered organic conductor $\alpha$-(BEDT-TTF)$_2$I$_3$[38, 39]. Figure 3a shows our experimental setup; the in-plane transverse ($x$-axis) Hall voltage is measured with applying the out-of-plane ($z$-axis) current, and the magnetic field of fixed strength is rotated within the $yz$-plane. In a simple metal, the Hall resistivity for such an experiment setup is given by $B_y/ne$, where $B_y = |\mathbf{B}|\sin\theta$ is the field component perpendicular to current, and $n$ is the carrier density. This leads the Hall resistivity $\rho_{zx}$ to follow a $\sin\theta$ dependence with the rotation of the field, which is indeed observed in YbMnBi$_2$ for weak fields (|**B**| < 1 T), as shown in Fig. 3b. However, $\rho_{zx}(\theta)$ starts to deviate from the $\sin\theta$ dependence for |**B**| > 2 T and such a deviation becomes significant

for **B** > 6 T and cusp-like peaks occur around $\theta = 90°$, as shown in Fig. 3c. Such unusual behaviors can be well understood in terms of the interlayer tunneling of the zeroth LLs' Weyl fermions. For the Weyl bands, when the energy spacing between the zeroth and first LL is greater than the LL's breadth, the DOS($E_F$) contributed by the Weyl points should monotonically increase upon increasing field and is proportional to the out-of-plane field component |**B**|$\cos\theta$. Therefore, a $\tan\theta$ dependence is expected for $\rho_{zx}(\theta)$ since $\rho_{zx} \propto B_y/ne \propto |\mathbf{B}|\sin\theta/|\mathbf{B}|\cos\theta = \tan\theta$. Indeed, we observed such a dependence, as shown in Fig. 3d, where $\rho_{zx}(\theta)$ is plotted against $\tan\theta$. It is interesting to note that $\rho_{zx}(\theta)$ measured at different fields collapse into a single line (i.e., the *black dashed line* in Fig. 3d) in a lower angle region, which is not surprising, since $\rho_{zx} \propto \tan\theta$ is field independent. At large angles, LL quantization is suppressed due to reduced $B_z$, causing the deviation from the $\tan\theta$ asymptote. The deviation angle is larger for higher fields, since the threshold field, $B_{c,z} = |\mathbf{B}|\cos\theta_c$, for the distinguishable zeroth LLs can be satisfied at higher angles.

Using the above model, which considers both the interlayer tunneling transport of the zeroth LLs and the Dirac bands' momentum relaxation transport, we can interpret the unusual angular dependence of Hall effect quantitatively. Our successful fits of $\rho_{zz}(\theta,\mathbf{B})$ shown in Fig. 2b, c suggests the assumption of negligible interband scattering is valid. Under this assumption, the total Hall conductivity $\sigma_{zx}^{\text{total}}$ can be expressed as

$$\sigma_{zx}^{\text{total}} = w_1 \cdot \sigma_{zx}^{\text{LL0}} + w_2 \cdot \sigma_{zx}^{\text{C}}, \qquad (3)$$

where $\sigma_{zx}^{\text{LL0}}$ and $\sigma_{zx}^{\text{C}}$ represent the Hall conductivities contributed by Weyl and Dirac bands, respectively. $w_1$ and $w_2$ represent the weight of the contribution for each type of band. For the zeroth LL's tunneling channel of the Weyl bands, its Hall conductivity $\sigma_{zx}^{\text{LL0}}$ under a field oriented at an angle of $\theta$ (Fig. 3a) can be expressed as

$$\sigma_{zx}^{\text{LL0}}(\mathbf{B}, \theta) = a\frac{B_y}{B_z^2}\exp\left(-b\frac{B_y^2}{B_z}\right), \qquad (4)$$

where $a$ and $b$ are material-dependent constants, $B_y = |\mathbf{B}|\sin\theta$, and $B_z = |\mathbf{B}|\cos\theta$. This equation holds when the zeroth LL is distinguishable from other LLs[38, 39]. The Hall conductivity of the momentum relaxation channel of the Dirac bands, $\sigma_{zx}^{\text{C}}$, can be found from the Boltzmann transport theory, i.e.,

$$\sigma_{zx}^{\text{C}}(\mathbf{B}, \theta) = \sigma_0\frac{\omega_c\tau}{1 + (\omega_c\tau)^2}, \qquad (5)$$

where $\sigma_0 = \frac{ne^2\tau}{m^*}$ and $\omega_c = \frac{eB\sin\theta}{m^*}$. The total Hall conductivity $\sigma_{zx}^{\text{total}}$ at the left side of Eq. (3) can be derived via taking the inverse of the resistivity tensor (see Methods), i.e.,

$$\sigma_{zx} = \frac{\rho_{yy}\rho_{xz}}{\rho_{xx}\rho_{yy}\rho_{zz} - \rho_{xy}\rho_{yx}\rho_{zz} - \rho_{xz}\rho_{zx}\rho_{yy}} \qquad (6)$$

where the resistivity tensor elements $\rho_{ij}$ ($i, j = x, y, z$) were directly obtained by measuring the voltage along the $+j$ direction with the current flowing along the $+i$ direction. With the measured resistivity tensor elements $\rho_{zx}$ (Fig. 3c), $\rho_{xx}$, $\rho_{yy}$, $\rho_{xy}$, $\rho_{yx}$, $\rho_{xz}$ and $\rho_{zz}$ (Supplementary Fig. 7a–e), we calculated the angular dependence of the total Hall conductivity $\sigma_{zx}^{\text{total}}(\theta)$ using Eq. (6). As shown in Fig. 3e, $\sigma_{zx}^{\text{total}}(\theta)$ displays a $\sin\theta$-like dependence at lower fields, but strongly deviates from it at high fields and a local minimum at $\theta = 90°$ gradually develops when the field is increased above 1 T. Such an unusual evolution of $\sigma_{zx}^{\text{total}}(\theta)$ with magnetic field cannot be described by the classical transport model, but can be understood by considering the zeroth LL quantum tunneling: at low fields, when the zeroth and first LLs

are not well separated, the quantum tunneling is minimized so that the total Hall conductivity is dominated by the classical momentum relaxation transport of Dirac bands (i.e., $\sigma_{zx}^{C}$). Thus, it exhibits a $\sin\theta$ dependence as predicted by Eq. (5), which can be approximated to $\sigma_{zx}^{C} \propto \omega\tau \propto |\mathbf{B}| \sin\theta$ for low fields. However, at high fields, the interlayer zeroth LL tunneling becomes important, thus leading to unusual angular dependence of $\sigma_{zx}^{total}$. The local minimum of $\sigma_{zx}^{total}$ at $\theta = 90°$ is caused by the suppression of the 2D LL quantization when the field is oriented close to the in-plane direction ($\theta = 90°$). This interpretation is verified by the quantitative fit of $\sigma_{zx}^{total}$ by Eq. (3), as shown in Fig. 3e.

In summary, we have studied the magnetotransport properties and their dependences on magnetic field orientation of Weyl semimetal YbMnBi$_2$. We find its $\rho_{xx}(B)$ exhibits remarkable SdH oscillations, from the analyses of which nontrivial Berry phases were extracted; this verifies the existence of Dirac band crossings above/below $E_F$. For AMR($\theta$) and $\sigma_{zx}(\theta)$, we observed unusual angular dependences under high fields. Both the AMR($\theta$) and $\sigma_{zx}(\theta)$ data can be well fitted by a model, which considers both the interlayer tunneling of Weyl fermions at the zeroth LLs and the momentum relaxation transport of other Dirac bands. Our finding highlights the unusual role of the zeroth LLs in transport, which is important to further understand the novel Dirac/Weyl fermion physics.

## Methods

**Single-crystal preparation.** The YbMnBi$_2$ single crystals were synthesized using a self-flux method with the stoichiometric mixture of Yb, Mn, and Bi elements. The starting materials were put into a small alumina crucible and sealed in a quartz tube in Argon gas atmosphere. The tube was then heated to 1050 °C for 2 days, followed by a subsequently cooling down to 400 °C at a rate of 3 °C h$^{-1}$. The plate-like single crystals as large as a few millimeters can be obtained. The composition and structure of these single crystals were checked using energy-dispersive X-ray spectroscopy and X-ray diffraction measurements.

**Magnetotransport and Hall effect measurements.** The magnetoresistance measurements were performed with a four-probe method. The low-field measurements are performed using a 9 T Physics Property Measurement System (PPMS, Quantum Design). The high-field measurements were conducted in the 31 T resistive magnet and the 45 T hybrid magnet at National High Magnetic Field Laboratory in Tallahassee.

The Hall resistivity $\rho_{ij}$ ($i \neq j$), including $\rho_{zx}$, $\rho_{xz}$, $\rho_{xy}$, and $\rho_{yx}$, were also measured using a four-probe method in PPMS. Due to slightly asymmetric electric contacts, a small but finite longitudinal component $\rho_{ii}$ is involved in each measured Hall (transverse) resistivity. Since the longitudinal component $\rho_{ii}$ follows $\rho_{ii}(\theta) = \rho_{ii}(360° - \theta)$, while the Hall resistivity follows $\rho_{ij}(\theta) = -\rho_{ij}(360° - \theta)$, $\rho_{ij}$ can be separated from $\rho_{ii}$ by symmetrizing the data: $\rho_{ij}(\theta) = [\rho_{ij}(\theta) - \rho_{ij}(360° - \theta)]/2$.

**Neutron-scattering measurements.** Single-crystal neutron diffraction measurements were performed on HB-3A four-circle diffractometer with the neutron wavelength $\lambda = 1.005$ Å at High Flux Isotope Reactor at Oak Ridge National Laboratory, and the data were refined with the FULLPROF[40].

**Determination of Berry phase for the Dirac bands.** In YbMnBi$_2$, the Dirac cones with the nodes located away from $E_F$ lead to the observed SdH oscillations with two major fundamental frequencies (Fig. 1d). For such multi-frequency oscillations, Berry phases cannot be obtained from the commonly used LL fan diagram, but can be determined through the direct fit of the oscillation pattern by the multiband LK formula[32], in which the observed SdH oscillations are treated as the linear superposition of several single-frequency oscillations. Each single-frequency oscillations can be described by the Lifshitz–Kosevich formula[30, 31], which takes Berry phase into account for a Dirac system[21]:

$$\frac{\rho_{osc}}{\rho(|\mathbf{B}| = 0)} = \frac{5}{2}\left(\frac{|\mathbf{B}|}{2F}\right)^{1/2}\sum_r \frac{1}{r^{1/2}} \frac{2r\pi^2 k_B T m^*/\hbar e|\mathbf{B}|}{\sinh(2r\pi^2 k_B T m^*/\hbar e|\mathbf{B}|)} e^{2r\pi^2 k_B T_D m^*/\hbar e|\mathbf{B}|}$$
$$\cos\left\{2\pi\left[\left(\frac{F}{|\mathbf{B}|} + \gamma\right)r - \delta\right]\right\} \quad (7)$$

where $r = 1,2,3,\ldots$ is the harmonic factor and $T_D$ is Dingle temperature. $\gamma = \frac{1}{2} - \frac{\phi_B}{2\pi}$ and $\phi_B$ is Berry phase; $\delta = 0$ and $\pm 1/8$ for the 2D and 3D systems, respectively. In our fits (Fig. 1e), the oscillation frequencies $F$ and the effective masses $m^*$ for each band are taken as known parameters, obtained from the analyses shown in the *inset* of Fig. 1d and Supplementary Fig. 4.

**Conductivity and resistivity tensor conversion.** In a 3D material, the resistivity ($\hat{\rho}$) and conductivity ($\hat{\sigma}$) tensors can be expressed as

$$\hat{\rho} = \begin{bmatrix} \rho_{xx} & \rho_{xy} & \rho_{xz} \\ \rho_{yx} & \rho_{yy} & \rho_{yz} \\ \rho_{zx} & \rho_{zy} & \rho_{zz} \end{bmatrix} \text{ and } \hat{\sigma} = \begin{bmatrix} \sigma_{xx} & \sigma_{xy} & \sigma_{xz} \\ \sigma_{yx} & \sigma_{yy} & \sigma_{yz} \\ \sigma_{zx} & \sigma_{zy} & \sigma_{zz} \end{bmatrix}. \quad (8)$$

The conductivity tensor can be obtained by taking the inverse of the resistivity tensor:

$$\hat{\sigma} = \hat{\rho}^{-1} = \frac{1}{\det(\hat{\rho})}\text{adj}(\hat{\rho}) \quad (9)$$

For the magnetic field within the $y$–$z$ plane, i.e., $\mathbf{B} = (0, B\sin\theta, B\cos\theta)$, the resistivity tensor elements $\rho_{yz}$, $\rho_{zy}$, $\rho_{xy}$, $\rho_{yx}$, $\rho_{xz}$, and $\rho_{zx}$ are expected to have the following relations: $\rho_{yz} = \rho_{zy} = 0$, $\rho_{xy} = -\rho_{yx}$, and $\rho_{xz} = -\rho_{zx}$. The first relationship of $\rho_{yz} = \rho_{zy} = 0$ is obvious and need not be verified. The latter two relations were verified with additional measurements, as shown in Supplementary Fig. 7c–e. By taking the inverse of the resistivity tensor Eq. (9), the conductivity tensor element $\sigma_{zx}$ can be derived as expressed in Eq. (6), while $\sigma_{zz}$ can be derived as

$$\sigma_{zz} = \frac{\rho_{xx}\rho_{yy} - \rho_{xy}\rho_{yx}}{\rho_{xx}\rho_{yy}\rho_{zz} - \rho_{xy}\rho_{yx}\rho_{zz} - \rho_{xz}\rho_{zx}\rho_{yy}} \quad (10)$$

**Data availability.** The authors declare that the main data supporting the findings of this study are available within this article and its Supplementary Information. Extra data are available from the corresponding author upon reasonable request. See author contributions for specific data sets.

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

## Acknowledgements

We thank the informative discussions with Sergey Borisenko. The work at Tulane is supported by the U.S. Department of Energy under EPSCoR Grant No. DE-SC0012432 with additional support from the Louisiana Board of Regents (support for personnel, materials, and travel). The work at the National High Magnetic Field Laboratory is supported by the NSF grant No. DMR-1206267, the NSF Cooperative Agreement No. DMR-1157490, and the State of Florida (support for high-field measurements). The work at UCR is supported by DOE BES Division under grant no. ER 46940-DE-SC0010597 (partial support for high-field measurements). X.K. acknowledges the start-up funds from Michigan State University. The neutron-scattering experiment at ORNL was sponsored by the Scientific User Facilities Division, Office of Science, Basic Energy Sciences, U.S. Department of Energy.

## Author contributions

The single crystals used in this study were synthesized and characterized by J.Y.L. The magnetotransport and Hall measurements in PPMS were carried out by J.Y.L., S.M.A.R., and L.S. The high-field measurements at NHMFL were conducted by J.Y.L., D.G., Y.S., S.C., and C.N.L. J.H. analyzed the data and wrote the manuscript with the input from J.Y.L. and Z.Q.M. T.Z., M.Z., X.L.K.., and H.B.C. performed neutron-scattering experiments and data analyses. All authors read and commented on the manuscript. The project was supervised by J.H. and Z.Q.M.

## Additional information

**Competing interests:** The authors declare no competing financial interests.

