## [Peer Review File · Nature Communications]

Reviewers' comments:

Reviewer #1 (Remarks to the Author):

This manuscript reports studies on the transport properties of the relativistic fermions of the zeroth Landau Level (LL). The zeroth LL is one of the key signatures of Dirac/Weyl fermions, which is absent in conventional electron systems. Therefore, probing transport properties of Dirac/Weyl fermions on the zeroth LL is of fundamental importance to the understanding the physics of Dirac/Weyl fermions. However, it is a challenging task to probe the direct contribution of the zeroth LL's Dirac/Weyl fermions to transport properties, since the zeroth LL is locked to the Dirac/Weyl point. When the Dirac/Weyl point is not close to the Fermi level, there would be no way to detect the signatures of the zeroth LL's Dirac/Weyl fermions through transport measurements. Taking the advantage of the recently discovered time reversal symmetry breaking Weyl semimetal YbMnBi₂ which possesses Weyl nodes at the Fermi level, the authors successfully revealed the transport mechanism of the zeroth LL's fermions through angle-resolved magnetoresistance and Hall resistivity measurements. They found that both interlayer magnetoresistance and in-plane Hall resistivity exhibit unusual angular dependence due to the increased degeneracy of the zeroth LL as the applied field is rotated from the out-of-plane to the in-plane direction and these phenomena can be understood only in terms of transport of the zeroth LL's Weyl fermions, which takes place through interlayer tunneling and/or momentum relaxation. As far as I know, the results reported in this manuscript are novel and particularly interesting; although many three dimensional Dirac and Weyl semimetals have been realized experimentally in recent years, there has been no similar report on transport properties of the zeroth LL's fermions. The authors not only present the interesting data, but also give quantitative interpretation through the fit of the experimental data to the theoretical model. In particular, their finding of $\tan(\theta)$ dependence of Hall resistivity is truly exciting, which provides solid evidence for the unusual transport mechanism of the zeroth LL's fermions. Overall, this work makes significant progress in unraveling novel physics of Dirac/Weyl fermions.

I see some technical issues which could be better improved. A) Fig.1 may be too crowded and could be split into two figures. B) The language can be better polished. C) a summary section should be added. D) more refs should be added to the introduction regarding 3D Dirac materials and the relevant work on ZrTe₅ should be mentioned since it is an important Dirac material.

Reviewer #2 (Remarks to the Author):

The authors report the observation of magneto transport in YbMnBi₂, and attribute the observed behaviors to the zeroth LL and linear band crossings of topological semimetal. Recently, the type-II Weyl semimetal is one of growing fields. I do not know other published works on the electric transport in YbMnBi₂, so the manuscript has a potential impact to the community. However, I am confused about the claims by the authors. I suggest the authors to address them properly before a further consideration.

(1) The authors did not give a clear picture of the energy bands of the system. In Fig. 1d, they show a clear SdH oscillation at low temperatures. The quantum limit was not reached even when the magnetic field is above 30T. However, then they argued that the Fermi energy was right at the Dirac point, and the observed main result in Fig. 2c was from the zeroth Landau level, where the sharp peak is obvious only when the field is larger than 9T. This is very confusing. They stated that this system consisted of electron and hole Dirac bands, at the connection points of these pockets, cone-like dispersions of Bi layers with node being at the Fermi energy appear. They argued that the previous Dirac bands contributed to the SdH oscillations, while the latter states led to the sharp peak in the AMR. I think that they should give a clear picture of the energy bands and the reason why the Dirac bands did not influence the sharp peaks of the AMR and why the Fermi

energy is right at the nodes of Weyl cones (no measurement supports it in the manuscript).

(2) The authors did not distinguish the concept of Landau levels in 2D systems and Landau bands in 3D cases. For example, on line 47 of the first paragraph they stated that a direct probe of relativistic fermions is not feasible for 3D Dirac or Weyl semimetals, such as Cd₃As₂. However, it is not true that in 3D Dirac (Weyl) semimetals the quantum limit when only the lowest Landau band is occupied has been achieved in several recent experiments.

(3) The authors fitted the data of the Hall resistance with a tangent dependence at lower angles. At larger angles, they only argued that the deviation is due to the suppression of the Landau quantization. It would be good if the authors could also fit the data at larger angles. Also, there are many pockets in this system. Do others pockets affect the Hall resistance?

(4) The deviation between the SdHO data and the two-carriers model at $1/B=0.03$ is somewhat obvious. It seems that the deviation is not from higher order terms or Zeeman splitting. Can the authors give a better fit?

Reviewer #3 (Remarks to the Author):

In this manuscript, the authors have presented the in-plane and interlayer transport features for antiferromagnetic YbMnBi₂ hosting Weyl fermions. In addition to the SdH oscillations in in-plane resistivity, they measured the detailed angle dependence of interlayer magnetoresistance effects. From the latter dataset, they argue the possible contribution of zeroth Landau mode in the interlayer conduction. They systematically measured the transport properties and display them in a clear manner. However, in my opinion, the manuscript has several serious problems. First, all the results shown in Fig. 1 (magnetic structure, magnetoresistance, details of SdH oscillations) have been already reported in ref. [18]. So I could not find any new discoveries in the present data on the in-plane transport and neutron diffraction.

Second, the analyses on the data of interlayer transport are not consistent. Although the authors assume that the tunneling processes are dominant in the interlayer transport, the temperature profile of interlayer resistivity R_{zz} in Fig. 1c exhibits nice metallic behaviour over the entire temperature range. Furthermore, the interlayer conductivity based on the tunneling between the zero-mode Landau levels (Eq. 1) gives negative magnetoresistance effects. However, as shown in Fig. 2b, the observed magnetoresistance effects are positive irrespective of the field angle. These facts mean that the interlayer conduction for YbMnBi₂ should be dominated by the usual coherent transport of the carriers from some other bands and hence the tunneling effect of the zero-mode of Landau level should be negligible. In fact, in the second paragraph on page 9, they fit the AMR data using the conventional B^2 magnetoresistance effect, which also reproduces the experimental results. Because the fitted result is almost independent of the model of the zero-mode Landau level transport, the zero mode plays a minimal role in the total interlayer conduction. Therefore, I don't think the main conclusion of this manuscript is supported by the present data and analyses.

From the above reasons, I don't recommend the publication of this manuscript as it is. After the significant revisions, it may be suitable for publication in some specialised journal.

The followings are other comments to be considered.

1. On the line 182, the authors adopt the B-linear magnetoresistance for σ_c , because the Weyl fermions reach the quantum limit at sufficiently high field. However, as shown in Fig. 1, clear SdH oscillations were observed even around 40 T, indicating the quantum limit is not achieved for the Weyl fermions with the nodes away from the Fermi energy. Therefore, the conventional B^2 MR should be adopted for σ_c in the entire field range.

2. According to Ref. 2, YbMnBi₂ appears to host several Weyl points not only at Fermi energy but also away from Fermi energy. It would be very helpful for the readers' understanding if the authors present the schematic band structure associated with the Weyl points.

3. The authors should distinguish the words 'Dirac' and 'Weyl'. I guess 'Weyl' fermion would be appropriate for YbMnBi₂.

4. I would recommend the authors to refer to the studies reporting the interlayer resistivity for the related Dirac materials, e.g., for SrMnBi₂ (Y. J. Jo, et al., PRL 113, 156602 (2014)) and EuMnBi₂ (H. Masuda et al., Sci. Adv. 2. e1501117 (2016)).

Response to Reviewer #1's comments:

First, we thank Reviewer #1 for taking the time to review our manuscript. We also appreciate that the reviewer made suggestions to improve our manuscript. We have made point-to-point responses (in black) to the reviewer's comments (in red) below.

This manuscript reports studies on the transport properties of the relativistic fermions of the zeroth Landau Level (LL). The zeroth LL is one of the key signatures of Dirac/Weyl fermions, which is absent in conventional electron systems. Therefore, probing transport properties of Dirac/Weyl fermions on the zeroth LL is of fundamental importance to the understanding the physics of Dirac/Weyl fermions. However, it is a challenging task to probe the direct contribution of the zeroth LL's Dirac/Weyl fermions to transport properties, since the zeroth LL is locked to the Dirac/Weyl point. When the Dirac/Weyl point is not close to the Fermi level, there would be no way to detect the signatures of the zeroth LL's Dirac/Weyl fermions through transport measurements. Taking the advantage of the recently discovered time reversal symmetry breaking Weyl semimetal YbMnBi₂ which possesses Weyl nodes at the Fermi level, the authors successfully revealed the transport mechanism of the zeroth LL's fermions through angle-resolved magnetoresistance and Hall resistivity measurements. They found that both interlayer magnetoresistance and in-plane Hall resistivity exhibit unusual angular dependence due to the increased degeneracy of the zeroth LL as the applied field is rotated from the out-of-plane to the in-plane direction and these phenomena can be understood only in terms of transport of the zeroth LL's Weyl fermions, which takes place through interlayer tunneling and/or momentum relaxation. As far as I know, the results reported in this manuscript are novel and particularly interesting; although many three dimensional Dirac and Weyl semimetals have been realized experimentally in recent years, there has been no similar report on transport properties of the zeroth LL's fermions. The authors not only present the interesting data, but also give quantitative interpretation through the fit of the experimental data to the theoretical model. In particular, their finding of $\tan(\theta)$ dependence of Hall resistivity is truly exciting, which provides solid evidence for the unusual transport mechanism of the zeroth LL's fermions. Overall, this work makes significant progress in unraveling novel physics of Dirac/Weyl fermions.

We thank the referee for giving a nice summary of our work and his/her evaluation on the importance and novelty of our findings.

I see some technical issues which could be better improved.

We really appreciate these suggestions given by the referee, which have been very helpful in improving our manuscript. We have addressed these technical issues following the referee's suggestions:

A) Fig.1 may be too crowded and could be split into two figures.

We fully agree with the reviewer that Fig.1 looks crowded. Fig. 1 in our original manuscript shows the structural and electronic property characterizations of YbMnBi₂. Some of the data is indeed not directly relevant to the focus of this manuscript, i.e. the unusual quantum transport behavior caused by the zeroth Landau levels. To make our presentation more succinct and focused, we have revised Fig. 1 following the reviewer's suggestion. In the revised manuscript,

we have moved the less-relevant data, i.e. the magnetic structure, the temperature dependence of magnetic ordering parameter, the temperature dependences of the in-plane and out-of-the plane resistivities and the effective mass analyses based on the SdH oscillations to the Supplementary Information. The revised Fig. 1 is shown below; it includes only the in-plane magnetoresistivity and the SdH oscillation analyses. To make our discussions clear, we have also added the schematics of the electronic band structure of YbMnBi₂ determined by the ARPES experiments¹ following Reviewer 2 and 3' suggestions.

Fig. 1 (for Reviewer 1) (a) Schematic of YbMnBi₂'s Fermi surface determined by ARPES experiments¹. The red and blue pockets correspond to electron- and hole-like pockets, respectively. The black dots represent Weyl points. (b) Schematic of the linear band crossing for the electron- (cut#3) and hole-like (cut #2) pockets and the Weyl point (cut #1), also determined by ARPES experiments¹. (c) Schematic of Landau levels for three types of band crossing shown in (b) under high magnetic fields. (d) The in-plane magnetoresistivity $\Delta\rho_{xx}/\rho_{xx}$ as a function of magnetic field along the out-of-plane direction. Inset, the FFT spectra of the SdH oscillations in $\Delta\rho_{xx}/\rho_{xx}$. (e) The fits of SdH oscillations at 2K and 18K to the two-band LK formula.

B) The language can be better polished.

We thank the reviewer for pointing out this problem. We have carefully read and polished the manuscript.

C) a summary section should be added.

We thank the reviewer for this good suggestion. We have added a summary section to the revised manuscript, as shown below:

“In summary, we have studied the in-plane magnetoresistivity $\rho_{xx}(B)$, the magnetic field orientation dependences of the out-of-plane magnetoresistivity $\text{AMR}(\theta)$ and in-plane Hall resistance $R_{zx}(\theta)$ under various magnetic fields for Weyl semimetal YbMnBi_2 . $\rho_{xx}(B)$ exhibits remarkable SdH oscillations, from the analyses of which non-trivial Berry phases were extracted; this verifies the existence of Dirac band crossings above/below E_F . For $\text{AMR}(\theta)$ and $R_{zx}(\theta)$, we observed unusual angular dependences under high fields. Both the $\text{AMR}(\theta)$ and $R_{zx}(\theta)$ data can be well fitted to a model which considers both the interlayer tunneling of Weyl fermions at the zeroth LLs and the momentum relaxation transport of other Dirac bands. Our finding highlights the unusual role of the zeroth LLs in transport, which is important to further understand the novel Dirac/Weyl fermion physics”.

D) more refs should be added to the introduction regarding 3D Dirac materials and the relevant work on ZrTe_5 should be mentioned since it is an important Dirac material.

We apologize for missing some important references. We have added more relevant references, including those on ZrTe_5 , to the revised manuscript.

Response to Reviewer #2's comments:

The authors report the observation of magneto transport in YbMnBi_2 , and attribute the observed behaviors to the zeroth LL and linear band crossings of topological semimetal. Recently, the type-II Weyl semimetal is one of growing fields. I do not know other published works on the electric transport in YbMnBi_2 , so the manuscript has a potential impact to the community. However, I am confused about the claims by the authors. I suggest the authors to address them properly before a further consideration.

We thank Reviewer #2 for taking the time to review our manuscript. The focus of our manuscript is to show how the Weyl fermions on the zeroth Landau level (LL) of YbMnBi_2 cause unusual transport behavior via measuring the field orientation dependences of interlayer magnetoresistivity and in-plane Hall resistance. Since the zeroth LL is one of the key signatures unique to Dirac/Weyl fermions, revealing transport evidence of the zeroth LL is of fundamental importance. We appreciate that the reviewer has seen the potential impact of our work. We apologize for not making our claims clear enough to the reviewer in the original manuscript and appreciate the reviewer's insightful comments, which have been very helpful in improving our manuscript. We have now addressed all the issues raised by the reviewer and revised the manuscript accordingly. We have made point-to-point responses (in black) to the reviewer's comments (in red) in the following.

(1) The authors did not give a clear picture of the energy bands of the system. In Fig. 1d, they show a clear SdH oscillation at low temperatures. The quantum limit was not reached even when the magnetic field is above 30T. However, then they argued that the Fermi energy was right at the Dirac point, and the observed main result in Fig. 2c was from the zeroth Landau level, where the sharp peak is obvious only when the field is larger than 9T. This is very confusing. They stated that this system consisted of electron and hole Dirac bands, at the connection points of these pockets, cone-like dispersions of Bi layers with node being at the Fermi energy appear. They argued that the previous Dirac bands contributed to the SdH oscillations, while the latter

states led to the sharp peak in the AMR. I think that they should give a clear picture of the energy bands and the reason why the Dirac bands did not influence the sharp peaks of the AMR and why the Fermi energy is right at the nodes of Weyl cones (no measurement supports it in the manuscript).

Fig. 1 (for Reviewer 2) (a) Schematic of YbMnBi₂'s Fermi surface determined by ARPES experiments¹. The red and blue pockets correspond to electron- and hole-like pockets, respectively. The black dots represent Weyl points. (b) Schematic of the linear band crossing for the electron- (cut#3) and hole-like (cut #2) pockets and the Weyl point (cut #1), also determined by ARPES experiments¹. (c) Schematic of Landau levels for three types of band crossing shown in (b) under high magnetic fields. (d) The in-plane magnetoresistivity $\Delta\rho_{xx}/\rho_{xx}$ as a function of magnetic field along the out-of-plane direction. Inset, the FFT spectra of the SdH oscillations in $\Delta\rho_{xx}/\rho_{xx}$. (e) The fits of SdH oscillations at 2K and 18K to the two-band LK formula.

We thank the reviewer for bringing up these important issues. Clarifications of these issues are indeed critical to the understanding of our claim as the reviewer pointed out. We apologize for not giving a clear electronic band structure for YbMnBi₂ in our original manuscript, which we believe caused the reviewer's confusion. The electronic band structure of YbMnBi₂ has been determined by Borisenko *et al*¹ using ARPES experiments and first principle calculations. In Fig. 1 attached above (i.e. Fig. 1 in the revised manuscript), we show the schematic of the projected Fermi surface on the k_x - k_y plane determined by the ARPES experiments (Fig. 1a) and the band dispersions along several typical momentums (Fig. 1b). The Fermi surface consists of the Weyl points (denoted by the black dots in Fig. 1a) and the hole-like (marked in blue) and electron-like (red) pockets comprised of linear Dirac bands. The Weyl points appear at the momentum points where electron- and hole-like pockets are connected, which is a typical signature of type II Weyl semimetal as claimed by Borisenko *et al*¹. Besides the Weyl points

shown in Fig. 1a, the first principle calculations also predicted another four Weyl points at different momentum points and two Weyl loops; but these features were not observed in the ARPES experiments¹. Given such a multi-band electronic structure, the transport properties of YbMnBi₂ should be contributed by both the Weyl points and the hole- and electron-like pockets. Due to the fact that the probed Weyl nodes are at E_F , while the Dirac bands forming the hole- and electron-like pockets cross at energies above or below E_F as illustrated in Fig. 1b, the Weyl and Dirac fermions are expected to exhibit distinct magnetotransport behaviors, as explained below.

As shown in Fig. 1c, under magnetic fields, both the Weyl bands and Dirac bands split to quantized Landau levels (LLs), the energies of which can be expressed as $\varepsilon_n = \pm v_F \sqrt{2e\hbar B |n|}$ ($n=0, \pm 1, \pm 2, \dots$)¹⁸. The $n = 0$ level corresponds to the zeroth LL, which is always locked to the Weyl/Dirac node upon field sweep no matter whether the node is at E_F or not (Fig. 1c). For the Dirac bands, since their crossing points, i.e. the Dirac nodes, are away from E_F (Fig. 1c, panel #2 & #3), their LLs would successively pass through E_F upon increasing the magnetic field, thus resulting in oscillating density of state DOS at E_F , which is manifested in quantum oscillations in our measured in-plane magneto-resistivity (Fig. 1d and 1e). By contrast, the Weyl nodes shown in Fig. 1a are located at E_F , the zeroth LLs of the Weyl bands are pinned to E_F regardless of magnetic field strength (Fig. 1c, panel #1). Under this circumstance, the Weyl bands are *not* expected to show quantum oscillations, since no LL passes E_F upon increasing the field. Instead, DOS(E_F) monotonically increases due to the increase of the zeroth LL's degeneracy. In general, it is hard to observe such an effect in transport measurements in most topological materials due to their Dirac/Weyl nodes away from E_F and/or the complexity of electronic band structure.

However, YbMnBi₂ offers an excellent opportunity to probe unusual transport behavior caused by the zeroth LL of Weyl bands since its electronic structure is quasi-2D and it has Weyl nodes at E_F . The Weyl nodes at E_F results in the presence of the zeroth LL at E_F as indicated above and the quasi-2D electronic structure make the LL quantization highly depend on the field orientation. As a result, the rotation of high magnetic fields from the out-of-plane to the in-plane direction would lead to a remarkable decrease of the zeroth LL degeneracy of the Weyl bands at E_F due to the suppression of LL quantization. This accounts for the sharp AMR peak present at $B//ab$ at fields above 9T as well as the unusual angular dependence of Hall resistance as discussed in the manuscript.

Next, we will answer the reviewer's second question raised in Comment (1), i.e. *why the Dirac bands did not influence the sharp peaks of the AMR?*

As stated above, the Fermi surface of YbMnBi₂ consists of the Weyl points and the hole- and electron-like pockets. All of them should contribute to transport properties. In our analyses of AMR data in Fig. 2c, we have indeed considered all these contributions, as shown by the equation used for the fit, *i.e.*

$$\rho_{\pm}(B, \theta) = 1/[\sigma_r^{LL0}(B, \theta) + \sigma_c(B, \theta)], \quad (1).$$

In this equation, σ_c represents the conductivity contributed by the momentum relaxation process, while σ_r^{LL0} represents the tunneling conductivity of Weyl fermions at the zeroth LLs located at E_F . The contribution of the hole- and electron-like pockets formed by the Dirac bands is included

in σ_c . Based on the evolution of AMR from the $\sin^2\theta$ dependence at low fields (e.g. see the data collected at 0.1 T in Fig.2c in the manuscript) to the sharp peak at $\theta = 90^\circ$ above 9T (Fig. 2c), we can see how the hole- and electron-like pockets affect the AMR sharp peaks. At low fields, these pockets should make dominant contributions to the transport, since these pockets should have a much greater DOS(E_F) than the Weyl points. The observation of the $\sin^2\theta$ dependence of AMR at low fields implies that the contribution of the hole- and electron-like pockets to AMR follows the classic Lorentz effect for which the interlayer magnetoresistivity is proportional to B_z^2 [$= (B\sin\theta)^2$]. When the field is remarkably increased, the DOS(E_F) of the Weyl points should increase dramatically. This is because that the Weyl nodes are at E_F in YbMnBi₂, such that the quantum limit of Weyl bands should be reached under a relatively low magnetic field though we cannot tell the exact threshold field (note that the quantum limit of the Dirac bands cannot be reached near 30T as the reviewer said, since the quantum oscillation frequencies associated with the Dirac bands are high, ~ 115 and 162T). Near the quantum limit of the Weyl bands, the LL degeneracy would enhance significantly, thus resulting in significantly increased DOS(E_F) at the zeroth LL. The gradual deviation from the $\sin^2\theta$ dependence in AMR upon increasing field suggests that Weyl fermions at the zeroth LL play a more important role in interlayer transport under high fields. Our successful fit of the AMR data to eq. (1) in the manuscript further suggests that the zeroth LL's Weyl fermions contribute to the interlayer transport via a tunneling process. The inclusion of σ_c in the fit indicates that the contributions of Dirac bands to the interlayer transport cannot be neglected even above 9T where a sharp peak is observed in AMR. We acknowledge the statement regarding the quantum limit in our original manuscript is confusing as the reviewer pointed out. When we spoke of the quantum limit, we did not clearly indicate which bands reach the quantum limit. Again, we apologize for that.

In the revised manuscript, we have added most of the above discussions to clarify the electronic band structure and addressed how the Dirac bands affect the AMR.

In the above comments, the reviewer also asked **why the Fermi energy is right at the nodes of Weyl cones (no measurement supports it in the manuscript)?**

This is a result probed in previous ARPES experiments by Borisenko *et al*¹ (see Fig .2b in Borisenko's manuscript). Our observation of the unusual angular dependences of interlayer magnetoresistance and in-plane Hall resistance under high magnetic fields can be well understood in light of the fact that YbMnBi₂ has Weyl nodes at E_F ¹, as discussed in the manuscript. We have made this clear in the revised manuscript.

(2) The authors did not distinguish the concept of Landau levels in 2D systems and Landau bands in 3D cases. For example, on line 47 of the first paragraph they stated that a direct probe of relativistic fermions is not feasible for 3D Dirac or Weyl semimetals, such as Cd₃As₂. However, it is not true that in 3D Dirac (Weyl) semimetals the quantum limit when only the lowest Landau band is occupied has been achieved in several recent experiments.

We apologize again for the confusing presentation in our original manuscript and thank the reviewer for pointing out these problems. Here we first clarify the difference of LLs between 2D and 3D systems, then explain the statement we made in the original manuscript that confused the reviewer. The most significant difference in LLs between 2D and 3D systems is that the LLs in

2D systems are more dependent on field orientation than 3D systems. Specifically, given the formation of LLs originates from cyclotron motion within 2D planes, LLs of 2D systems are determined by the magnetic field component perpendicular to the 2D plane, B_z and the decrease of B_z leads to decreased LL degeneracy. LLs vanish when B_z decreases to zero. In contrast, in 3D systems, LLs appear for any magnetic field orientations and do not change with field orientation if the electronic structure is isotropic for a given system.

Our work reported in this manuscript takes advantage of the quasi-2D electronic structure of YbMnBi₂ as well as its Weyl nodes at E_F to probe transport properties of Weyl fermions at the zeroth LLs. With the quasi-2D electronic structure, the field rotation from the out-of-plane to in-plane direction would lead the LL degeneracy to decrease, thus resulting in the decrease of $\text{DOS}(E_F)$. Since the zeroth LL of Weyl bands is at E_F in YbMnBi₂ as shown in Fig. 1c (panel#1), our AMR and angle dependent Hall resistance can probe the variation of the $\text{DOS}(E_F)$ of the zeroth LL with the field rotation.

Our statement regarding quantum limit in the original manuscript is confusing as pointed out by the reviewer. The reviewer is right: the quantum limit can be reached at high magnetic fields for 3D Dirac and Weyl semimetals, such as Cd₃As₂^{2,3}, TaP⁴ and NbAs⁵. In our original manuscript, we made the statement: “Direct probe of such relativistic fermion transport is not feasible for recently-discovered, three dimensional (3D) Dirac/Weyl semimetals”. This statement is indeed very confusing. What we meant by this statement is that direct probe of the enhanced conductivity due to the increased degeneracy of the zeroth LL is not feasible though the quantum limit is reached. This is because that these materials have 3D electronic structures such that the field rotation might not lead to significant changes of LL degeneracy. In the revised manuscript, we have removed these confusing statements and added some of the above discussions to make our claim clear.

(3) The authors fitted the data of the Hall resistance with a tangent dependence at lower angles. At larger angles, they only argued that the deviation is due to the suppression of the Landau quantization. It would be good if the authors could also fit the data at larger angles. Also, there are many pockets in this system. Do others pockets affect the Hall resistance?

This is a good suggestion, which has motivated us to perform further analyses for the angular dependence of Hall resistance R_{zx} . We have now been able to fit the R_{zx} data within the whole angle range. This new result provides additional support for our claim. We thank the reviewer for giving this suggestion.

As indicated above, the Fermi surface of YbMnBi₂ consists of the type-II Weyl points and the electron- and hole-like pockets formed by the Dirac bands. The Dirac bands are expected to manifest themselves in quantum oscillations in magnetoresistivity and Hall resistivity, since their zeroth LLs are away from E_F , as shown in Fig. 1c (panels #2 and 3#). However, high field is needed to observe such quantum oscillations. As indicated above, the quantum oscillations associated with Dirac bands in YbMnBi₂ have been seen in the in-plane magnetoresistivity for $B > 15\text{T}$ (Fig. 1d), but were not observable up to 31 T in the out-of-plane magnetoresistivity (Fig. 2b in the manuscript). In our Hall resistance measurement, the available maximum field is only

9T, so we did not observe any quantum oscillations due to the Dirac bands. Therefore, within such a low field range, the contribution of Dirac bands to Hall resistance should follow the classical model, i.e., $R_{zx,D} \approx B_y/ne \propto B\sin\theta$, for the measurement setup shown in Fig. 2a attached below (i.e. Fig. 3a in the revised manuscript). This is indeed verified by the $\sin\theta$ dependence of R_{zx} seen at low fields where the Dirac bands make dominant contributions to transport as indicated above (see Fig. 2b below)

For the Weyl points, their contribution to the Hall resistance should come from the Weyl fermions at the zeroth LL as discussed above. Since the Weyl nodes are at E_F and the zeroth LL is pinned to E_F upon increasing magnetic field as shown in Fig. 1c (panel#1), the Weyl bands are not expected to show quantum oscillations. Given that YbMnBi₂ possesses a quasi-2D electronic structure and its zeroth LL's degeneracy $\propto B_z = B\cos\theta$, a rough but straightforward estimation leads to $R_{zx,W} \approx B_y/ne \propto B\sin\theta/B\cos\theta = \tan\theta$, which is indeed observed in the low angle range where the LL quantization is strong, as shown in Fig. 2c below (i.e. Fig. 3c in the manuscript). An explicit expression of Hall resistivity due to such a quantum tunneling effect of the zeroth LL has been theoretically derived as

$$\rho_{zx}^{LL0} = \frac{B_y}{B_z} \cdot \frac{a}{1 + (b \cdot B_z)^2}, \quad (2),$$

where a and b are material dependent constants⁶. As indicated in the manuscript, this equation is based on a theoretical model which assumes that the interlayer transport occurs through the tunneling process of relativistic fermions at the zeroth LL. From eq. (2), it can be seen that remarkable quantum effect should appear at high magnetic fields oriented at low angles where the zeroth LL is distinguishable from other LLs^{6,7}. At lower fields or high fields oriented along larger angles where LL quantization is strongly suppressed, the angular dependence of R_{zx} resumes to the classical scenario, i.e. $R_{zx} \approx B_y/ne \propto B\sin\theta$.

We can combine the contributions from the Dirac and Weyl bands described above by assuming their contributions toward Hall conductivity are additive, i.e.

$$\sigma_{zx}^{total} = w_1 \cdot \sigma_{zx}^{LL0} + w_2 \cdot \sigma_{zx}^C, \quad (3),$$

where σ_{zx}^{total} is the total Hall conductivity, σ_{zx}^{LL0} and σ_{zx}^C represent the Hall conductivity contributed by the Weyl and Dirac bands, respectively. w_1 and w_2 in eq. (3) represents the weight of the contribution for each type of band. To fit the measured angular dependence of Hall resistance (Fig. 2d below) to eq. (3), we convert the conductivities in eq. (3) to the resistivities. Thus eq. (3) can be expressed as:

$$\frac{1}{\rho_{zx}^{total}} = w_1 \cdot \frac{1}{\rho_{zx}^{LL0}} + w_2 \cdot \frac{1}{\rho_{zx}^C}, \quad (4)$$

As stated above, the Hall resistivity for the Dirac bands should follow the classical model: $\rho_{zx}^C = B_y/ne \propto B_y$, while the Hall resistivity for the Weyl bands is dependent on the zeroth LL's degeneracy and can be expressed as eq. (2). Using this simplified model, we can fit the angular dependence of ρ_{zx}^{total} in the full angular range (0 - 180°) under various magnetic fields (0.5 - 9T). The weight factors w_1 and w_2 are set to be field-dependent in these fits, while the parameters a and b are set to be identical for different fields since they are material constants. As shown in Fig. 2d below (i.e. Fig. 3d in the revised manuscript), this model, i.e. eq. (4), reproduces the

measured data very well. These fitting results further support our claim that the zeroth LLs of the Weyl bands participate in the interlayer transport through quantum tunneling.

Fig. 2 (for reviewer) (a) Experimental setup for Hall effect measurements. (b) Angular dependence of Hall resistance R_{xx} at $B=0.5\text{T}$ and $T=2\text{K}$, which follows a $\sin\theta$ dependence as indicated by the fitted solid curve. (c) R_{xx} plotted against $\tan\theta$; the $\tan\theta$ asymptote (*i.e.* the dashed line) can be observed at low angles. (d) Angular dependence of R_{xx} at $T=2\text{K}$ under different magnetic fields. Data at different fields have been shifted for clarity. The solid lines represent the fits to eq. (4). Inset: the weight ratio w_1/w_2 at different fields.

Furthermore, from the fits described above, we can extract the weight ratio w_1/w_2 as depicted in the inset of Fig. 2d (*i.e.* Fig. 3d in the current manuscript), which provides the information on how the quantum tunneling transport of the zeroth LL's Weyl fermions competes with the Dirac fermion transport through momentum relaxation in YbMnBi_2 . At low fields (e.g. $B = 0.5\text{T}$), $w_1/w_2 = 0$, indicating that the Hall effect is fully classical since LL quantization is weak. Increasing the magnetic field leads to the increase of the LL spacing. When the zeroth LL of the Weyl bands become gradually distinguishable from the other LLs, w_1/w_2 also gradually increases and reaches 1.25 at $B = 9\text{T}$, indicating that interlayer tunneling of Weyl fermions at the zeroth LL become significant.

One may ask why the interlayer Dirac fermion transport occur through the momentum relaxation process in YbMnBi_2 , while the zeroth LLs' Weyl fermions transport takes place via the interlayer tunneling. This may be attributed to the different dimensionality of the Dirac and Weyl bands. The band structure studies by Borisenko *et al.*¹ suggest the Weyl bands are much more 2D-like in YbMnBi_2 and the Weyl cone is highly anisotropic, with its Fermi velocity V_F along the k_z direction being two orders of magnitude less than V_F along the k_x direction. Under this circumstance, it is reasonable to assume the interlayer quantum tunneling of the zeroth LLs' Weyl fermions. However, the Dirac bands should have higher dimensionality, which is evidenced by the fact that YbMnBi_2 exhibits a moderate electronic anisotropy as reflected in the ρ_{zz}/ρ_{xx} resistivity ratio (~ 36 at $T = 2\text{K}$). If the Dirac bands were also highly 2D-like as the Weyl

bands are, a large electronic anisotropy would be expected, inconsistent with the experimental observation. Therefore, it is reasonable to assume that the Dirac bands contribute to the interlayer transport through momentum relaxation (i.e. coherent band transport).

We have added the above discussions and new data analyses to the revised manuscript.

(4) The deviation between the SdHO data and the two-carriers model at $1/B=0.03$ is somewhat obvious. It seems that the deviation is not from higher order terms or Zeeman splitting. Can the authors give a better fit?

Following the reviewer's suggestion, we have made further efforts to improve the fits of the SdHO data to the two-carriers model. In our previous fits, only fundamental frequencies F_α and F_β are included in the fits. The higher harmonic components $2F_\beta$ and $3F_\beta$ revealed in the Fourier transform (inset to Fig.1d in the current manuscript) were not included in the fits. When these components as well as the $4F_\beta$ component (which is very weak and not shown in the FFT spectra in the inset to Fig.1d) are included in the fits, the deviation from fit becomes much smaller, as shown in Fig. 1e in the current manuscript. The Berry phases extracted from the fits show only very small changes after adding the high harmonic components to the fits. Again, we thank the reviewer for bringing up this issue. We have added the improved fit to the revised manuscript.

Response to Reviewer #3's comments:

In this manuscript, the authors have presented the in-plane and interlayer transport features for antiferromagnetic YbMnBi₂ hosting Weyl fermions. In addition to the SdH oscillations in in-plane resistivity, they measured the detailed angle dependence of interlayer magnetoresistance effects. From the latter dataset, they argue the possible contribution of zeroth Landau mode in the interlayer conduction. They systematically measured the transport properties and display them in a clear manner.

We thank the reviewer for reviewing our manuscript. We also appreciate the questions/criticisms raised by this reviewer, which have been very helpful in improving our manuscript. We have now addressed all of the reviewer's questions/criticisms and revised the manuscript accordingly. In the following, we have made point-to-point responses (in black) to the reviewer's comments (in red). The figures (for reviewer) refers to the figures in this response letter.

However, in my opinion, the manuscript has several serious problems. First, all the results shown in Fig. 1 (magnetic structure, magnetoresistance, details of SdH oscillations) have been already reported in ref. [18]. So I could not find any new discoveries in the present data on the in-plane transport and neutron diffraction.

We agree with the reviewer that the neutron scattering and SdH oscillations in in-plane transport data have been reported by Wang *et al.*⁸ We have clearly indicated it in the manuscript and quoted the relevant reference. However, since our magnetoresistance measurements were made to much higher fields, up to 45 T, the SdH oscillations probed in our experiments revealed new important information which was not seen in Wang *et al.*'s experiments⁸ which were made only

up to 35T. We apologize for not clearly indicating what are our new discoveries in our SdH data. We thank the referee for seeing this problem.

Because of the limited field range, Wang *et al.*⁸ missed observing the secondary oscillation frequency component in their Fast Fourier transform (FFT) analyses of the SdH oscillations of the in-plane magnetoresistivity ρ_{xx} . In fact, their $\rho_{xx}(B)$ data exhibit a clear feature which implies multiple oscillation frequencies even below 31T; that is, the oscillation peak between valley 6 and 7 is strongly suppressed as compared to other peaks, as shown in Fig. 3b in their manuscript. However, their FFT spectrum of the SdH oscillations in $\rho_{xx}(B)$ shows only a broad peak at about 130T. Clearly this FFT spectrum has large uncertainty and the second frequency component was not resolved due to the limited field range. This is also the reason why their oscillation frequencies probed in their Hall resistivity ρ_{xy} (108T and 164 T) are inconsistent with that probed in ρ_{xx} . Therefore, the Berry phase they extracted using the Landau index fan diagram based on the SdH oscillations of ρ_{xx} has a large error due to the presence of multiple frequencies. Indeed, their determined Berry phase is 0.42π , much less than the expected value of π . They also tried extracting the Berry phase from the fan diagram based on the SdH oscillation in ρ_{xy} . Again, the fan diagram method is not effective for the SdH oscillations with multiple frequencies, since the interference of the oscillations with different frequencies does not allow for precise determination of the peak and valley positions of the SdH oscillations⁹.

By contrast, our observed SdH oscillations in ρ_{xx} , which were measured up to 45T, show clear features arising from multiple oscillation frequencies (see Fig. 1d attached below). Our FFT analyses indeed show two frequencies at $F_\alpha=115\text{T}$ and $F_\beta=162\text{T}$ (inset to Fig. 1d). To extract the Berry phases accumulated in cyclotron orbits associated with these two frequencies, we fitted the SdH oscillations in ρ_{xx} with a two-band Lifshitz-Kosevich (LK) formula which takes Berry phase into account for a topological fermion system. This approach has been shown to be particularly effective for finding Berry phases in a topological material with multiple bands⁹. As seen in Fig. 1e (attached below for reviewer), our SdH oscillation data at 2K and 18K can be nicely fitted with a two-band LK formula when the higher harmonic components of F_β (i.e. $2F_\beta$, $3F_\beta$ and $4F_\beta$) are taken into account (see the Methods section for more details) (note that the $4F_\beta$ component is very weak and not shown in the FFT spectra in the inset to Fig.1d)). The Berry phases derived from these fits are 0.8π for F_α and -0.6π for F_β , which are clearly non-trivial. The fits in our original manuscript show some deviations, which are due to the fact that we did not consider higher harmonic components for F_β . YbMnBi₂ is a multiple-band system and its Fermi surface consists of the Weyl points and the hole- and electron-like pockets formed by the Dirac bands (see below for more details). The SdH oscillations observed in ρ_{xx} should arise from the Dirac bands as discussed in the manuscript. Our precise determination of non-trivial Berry phase verifies that the hole/electron-like pocket is indeed comprised of topologically non-trivial bands, *i.e.* the Dirac bands.

We have revised Fig. 1 to highlight our new discoveries in the revised manuscript. The current version of Fig. 1 (attached below) includes the in-plane magnetoresistivity data, the FFT analyses of SdH oscillations as well as the fits of the SdH oscillations to the two-band LK formula. We have moved the neutron scattering data as well as the effective mass analyses to the Supplementary Information. Additionally, we have added a schematic of the electronic band

structure determined by previous ARPES experiments ¹ to revised Fig. 1 following Reviewer 3 and 2's suggestions. The addition of this schematic band structure makes our discussions much clearer.

We would like to emphasize the central point of this manuscript is that we have demonstrated the unusual interlayer transport of the zeroth LL's Weyl fermions, which has never been reported in any other topological semimetals. Our demonstrations are based on the AMR data shown in Fig. 2 and the angle-dependent Hall resistivity data in Fig. 3. These data are novel and have never been reported.

Fig. 1 (for Reviewer 3) (a) Schematic of YbMnBi₂'s Fermi surface determined by ARPES experiments ¹. The red and blue pockets correspond to electron- and hole-like pockets, respectively. The black dots represent Weyl points. (b) Schematic of the linear band crossing for the electron- (cut#3) and hole-like (cut #2) pockets and the Weyl point (cut #1), also determined by ARPES experiments ¹. (c) Schematic of Landau levels for three types of band crossings shown in (b) under high magnetic fields. (d) The in-plane magnetoresistivity $\Delta\rho_{xx}/\rho_{xx}$ as a function of magnetic field along the out-of-plane direction. Inset, the FFT spectra of the SdH oscillations in $\Delta\rho_{xx}/\rho_{xx}$. (e) The fits of SdH oscillations at 2K and 18K to the two-band LK formula (see the method section for more details for the fits).

Second, the analyses on the data of interlayer transport are not consistent. Although the authors assume that the tunneling processes are dominant in the interlayer transport, the temperature profile of interlayer resistivity R_{zz} in Fig. 1c exhibits nice metallic behaviour over the entire temperature range. Furthermore, the interlayer conductivity based on the tunneling between the zero-mode Landau levels (Eq. 1) gives negative magnetoresistance effects. However, as shown in Fig. 2b, the observed magnetoresistance effects are positive irrespective of the field angle.

These facts mean that the interlayer conduction for YbMnBi₂ should be dominated by the usual coherent transport of the carriers from some other bands and hence the tunneling effect of the zero-mode of Landau level should be negligible. In fact, in the second paragraph on page 9, they fit the AMR data using the conventional B^2 magnetoresistance effect, which also reproduces the experimental results. Because the fitted result is almost independent of the model of the zero-mode Landau level transport, the zero mode plays a minimal role in the total interlayer conduction. Therefore, I don't think the main conclusion of this manuscript is supported by the present data and analyses.

The reviewer raised several issues in the above comments. We believe they were caused by our insufficient/unclear discussions in the original manuscript. We will address them one by one in the following.

(1) Metallic interlayer transport and tunneling process.

Quantum tunneling is due to the barrier penetration of particles. The tunneling of charge carriers appears wherever the barrier exists. YbMnBi₂ possesses a layered structure with the 2D Bi square-net planes being separated by the Yb-MnBi₄-Yb layers. The first principle calculations¹ have revealed that the conduction electrons in this material are mostly Bi 6*p*-electrons from the 2D Bi planes and its electronic structure is quasi-2D, which has indeed been demonstrated by ARPES experiments¹. The Yb-MnBi₄-Yb layers in between the neighboring 2D Bi planes should act as barriers for the interlayer transport, hence the tunneling process between the neighboring 2D Bi planes is naturally expected.

It is not surprising to observe a metallic temperature dependence when tunneling is present. Here, we would like to give a more specific example. One of the corresponding authors of this manuscript, Zhiqiang Mao, was previously involved in the tunneling studies on the spin-triplet superconductor. He and his co-workers successfully probed Andreev surface bound states in the 3-K superconducting phase of Sr₂RuO₄ using cleaved junctions (Mao et al., PRL 87, 037003 (2001)¹⁰). The junctions were made by tightly fixing two cleaved crystals with a teflon frame. Although the junction resistance as a function of temperature shows metallic behavior in the whole temperature range (see Fig. 1 in Ref.¹⁰), significant tunneling behavior was observed, which enabled the observation of the Andreev bound state in Ru-Sr₂RuO₄¹⁰.

Generally, in a material with layered structure, it has been well-established that coherent transport through momentum relaxation and tunneling process coexist in the interlayer transport¹¹. As pointed out in Ref. 11 and references therein, depending on the relative weight and specific parameters of these two transport channels, the interlayer resistivity can exhibit various types of temperature dependences¹¹, including purely metallic (*e.g.* overdoped high- T_c cuprates¹²), insulating-like (*e.g.* TaS₂¹³), non-monotonic with a maximum at a certain temperature (*e.g.* Sr₂RuO₄¹⁴, κ -(BEDT-TTF)₂-Cu(SCN)₂¹⁵, (Bi_{0.5}Pb_{0.5})₂Ba₃Co₂O₇ and NaCo₂O₄¹⁶) and non-monotonic with a minimum at a certain temperature (*e.g.* underdoped cuprates¹²), *etc.*

(2) Why does $\rho_{zz}(B)$ show positive magnetoresistance ?

The referee is right: negative MR is indeed expected for the tunneling transport between neighboring 2D Bi planes, as reflected in the tunneling model used for fitting our data in the manuscript, i.e.

$$\sigma_i^{LL0} = A \cdot |B \cos \theta| \exp\left[-\frac{1}{2} \frac{ed^2 (B \sin \theta)^2}{\hbar |B \cos \theta|}\right], \quad (1)$$

which was adopted from Osada et al, JPSJ, 77, 084711¹⁷. However, as have been stated in the manuscript, the coherent momentum relaxation transport should coexist with the interlayer quantum tunneling transport in YbMnBi₂ due to its finite interlayer coupling and multiple-band characteristic. The positive MR seen in our experiments is due to the competition between the negative MR component from the tunneling channel and the positive MR component from the momentum relaxation channel.

Next we first clarify the electronic band structure of YbMnBi₂, which was not made clear in the original manuscript, and then explain why the interlayer magnetoresistance observed in our experiments is positive despite the interlayer tunneling of the zeroth LL mode. As shown in Fig. 1a, the Fermi surface of YbMnBi₂ consists of the Weyl points and the hole-like (marked in blue) and electron-like (red) pockets comprised of the Dirac bands¹. Given such a multi-band electronic structure, the transport properties of YbMnBi₂ should be contributed by both the Weyl and Dirac bands.

As shown in Fig. 1c, under magnetic fields, both the Weyl bands and Dirac bands split to quantized Landau levels (LLs), the energies of which can be expressed as $\varepsilon_n = \pm v_F \sqrt{2e\hbar B |n|}$ ($n=0, \pm 1, \pm 2, \dots$)¹⁸. The $n = 0$ level corresponds to the zeroth LL, which is always locked to the Weyl/Dirac node upon field sweep no matter whether the node is at E_F or not, as shown in Fig. 1c attached above. For the Dirac bands, since their crossing points, i.e. the Dirac nodes, are away from E_F (Fig. 1c, panel #2 & #3), their $n \neq 0$ LLs would successively pass through E_F upon increasing the magnetic field, thus resulting in oscillating density of state DOS at E_F , which is manifested in quantum oscillations in our measured in-plane magnetoresistivity (Fig. 1d and 1e). By contrast, since the Weyl nodes probed by ARPES¹ are located at E_F , the Weyl bands' zeroth LLs are pinned to E_F regardless of magnetic field strength (Fig. 1c, panel #1). Therefore, the Weyl bands are *not* expected to show quantum oscillations, since no LLs pass E_F upon increasing the field. Instead, $\text{DOS}(E_F)$ monotonically increases due to the increase of the zeroth LL's degeneracy.

In our model used for the data analyses in Fig. 2 and 3, we assume the interlayer Dirac fermion transport occurs through the momentum relaxation process (*i.e.* coherent band transport), while the zeroth LLs' Weyl fermions transport takes place via interlayer tunneling. This assumption is based on the different dimensionality of the Dirac and Weyl bands. The band structure studies by Borisenko *et al.*¹ suggest the Weyl bands are highly 2D-like in YbMnBi₂. However, the Dirac bands should have higher dimensionality, which is evidenced by the fact that YbMnBi₂ exhibits a moderate electronic anisotropy as reflected in the ρ_{zz}/ρ_{xx} resistivity ratio (~ 36 at $T = 2$ K). If the Dirac bands were also highly 2D-like, a large electronic anisotropy would be expected, inconsistent with the experimental observation. Therefore, it is reasonable to assume that the Dirac bands contribute to the interlayer transport through momentum relaxation, while the zeroth

LLs of the Weyl bands contribute to the interlayer transport through tunneling. The former channel has positive magnetoresistance (MR), but the latter one has negative MR.

Next, we will discuss how the tunneling channel leads to unusual $\rho_{zz}(B)$. As shown in Fig. 2b in the revised manuscript, when the field is aligned within the plane (*i.e.* $\theta = 90^\circ$) where LLs quantization vanishes, $\rho_{zz}(B)$ exhibits B^2 dependence in a low field region, but gradually evolves to a linear field dependence above 10T. However, as the field is tilted toward the out-of-plane direction ($\theta = 0^\circ$) where LL quantization develops, a sub-linear field dependence in $\rho_{zz}(B)$ emerges. Such an unusual evolution of MR with θ cannot be understood in light of the MR induced by the classical orbital effect or by other quantum effects such as weak anti-localization, but can be understood in terms of the gradually developed negative MR in the tunneling channel. Such negative MR component competes with the positive MR component arising from the momentum relaxation channel of the Dirac bands, resulting in the unusual sub-linear field dependence in $\rho_{zz}(B)$ for $\theta < 90^\circ$. This argument is quantitatively supported by the nice fits of $\rho_{zz}(B)$ (Fig. 2b) and $\rho_{zz}(\theta)$ (Fig. 2c) to the following equation for $\theta < 90^\circ$,

$$\rho_{zz}(B, \theta) = 1/[\sigma_t^{LL0}(B, \theta) + \sigma_c(B, \theta)], \quad (2),$$

where σ_t^{LL0} [*i.e.* eq. (1)] and σ_c represent the conductivities of tunneling and momentum relaxation channels respectively. Our claim is further supported by the fits of the unusual angular dependences of Hall resistance R_{zx} to the same model, as discussed in the manuscript.

(3) The role of zeroth LL in the interlayer transport

In our original manuscript, we stated “the conduction through the momentum relaxation channel alone can also account for the AMR data” in Fig. 2c. Indeed, this is a confusing and incorrect statement, which led the reviewer to think that the zeroth LL mode plays a minimal role in interlayer transport. When we used the model of momentum relaxation alone to fit our AMR data, one assumption was made: the interlayer transport at high fields is still dominated by the zeroth LL’s Weyl fermions and the $DOS(E_F)$ contributed by the Weyl points varies with the field and its orientation. In other words, the zeroth LL still plays a critical role in this model. We apologize for not making this clear in the original manuscript.

However, after reading the reviewer’s comments, we have carefully inspected the momentum relaxation-alone model and find this model is indeed inapplicable in interpreting our data, as discussed below. In this model, we considered a pure momentum relaxation process for the Weyl point states. The $DOS(E_F)$ contributed by the Weyl points is determined by the zeroth LL’s degeneracy since the zeroth LL is pinned to the E_F . Given the 2D LL quantization in YbMnBi₂, $DOS(E_F) \propto B_z$ (the magnetic field component perpendicular to the plane). Therefore, by assuming $\sigma \propto DOS(E_F)$, a resistance peak is naturally expected for $\theta = 90^\circ$. However, the assumption of $\sigma \propto DOS(E_F)$ is oversimplified and neglects the influence of the scattering rate on conduction. From the well-established transport theory^{19,20}, $\sigma \propto \tau$ and the scattering probability $1/\tau$ varies with the number of available states that electrons can be scattered into (*i.e.* $1/\tau \propto DOS(E_F)$)^{31,32}. This leads to $\sigma \propto 1/DOS(E_F)$ rather than $\propto DOS(E_F)$. Therefore, our original analyses using a pure momentum relaxation model is incorrect. So the zeroth LLs’ tunneling of the Weyl points is the only possible mechanism for the interpretation of unusual interlayer

transport observed in our experiments. We apologize for making this mistake. We have removed the discussions on the momentum relaxation-alone model in the revised manuscript.

Additionally, the role of the zeroth LL in transport have also been supported by the angular dependence of Hall resistance. As stated above, when LL quantization is weak at low fields (e.g., $B=0.5\text{T}$), the angular dependence of Hall resistance follows a classic $\sin\theta$ dependence. However, when LL quantization is strong enough to separate the zeroth LL from other LLs, Hall resistance exhibits unusual angular dependence with a cusp-like peak showing up at $\theta=90^\circ$ (Fig. 3d in the revised manuscript). Such an angular dependence can be understood in terms of the contributions from the zeroth LL. *In the revised manuscript, we have performed further analyses and successfully fit the angular dependences of Hall resistance in the full angular range (0-180°) under different fields (0.5-9T) using a model which considers both the tunneling of the zeroth's LLs of the Weyl points and the coherent momentum relaxation of Dirac fermions (see Fig. 3d in the revised manuscript).*

We thank the reviewer for giving the criticisms in the above comments, which have been very helpful for us to improve the discussions in the revised manuscript.

From the above reasons, I don't recommend the publication of this manuscript as it is. After the significant revisions, it may be suitable for publication in some specialised journal.

We hope we have convinced the reviewer our claim is well supported by our data. Given the zeroth LL is one of the key signatures of topological bands, finding transport evidence for the zeroth LL for the first time reversal symmetry breaking Weyl semimetal YbMnBi₂ is of fundamental importance. We also hope the reviewer has been convinced that our work has broad interests and fits into the scope of Nature Communications.

The followings are other comments to be considered.

1. On the line 182, the authors adopt the B-linear magnetoresistance for σ_c , because the Weyl fermions reach the quantum limit at sufficiently high field. However, as shown in Fig. 1, clear SdH oscillations were observed even around 40 T, indicating the quantum limit is not achieved for the Weyl fermions with the nodes away from the Fermi energy. Therefore, the conventional B^2 MR should be adopted for σ_c in the entire field range.

We thank the reviewer for pointing out this issue and apologize for not providing clear justifications for the fit of magnetoresistivity in the original manuscript.

As indicated above, the Fermi surface of YbMnBi₂ consists of not only the Weyl points, but also the electron- and hole-like pockets formed by the Dirac bands (Fig. 1a); the quantum oscillations shown in Fig. 1d should arise from the Dirac bands. As pointed out by the reviewer, the quantum limit for these Dirac bands are clearly not reached around a field of 40T. Our statement of reaching a quantum limit in the original manuscript refers to the quantum limit of the Weyl bands. This was not made clear in the original manuscript, which caused some confusions. Since the Weyl nodes are at E_F , the quantum limit of the Weyl bands should be reached under a relatively low field though we cannot tell its exact threshold value.

Although linear MR is expected to occur at quantum limit theoretically²¹, it is worth noting that linear MR has also been widely observed in many materials at low fields when quantum limit is not reached, such as $\text{Ag}_{2+\delta}\text{Se}$ ²², LaAgSb_2 ^{23,24}, Cd_3As_2 ^{2,25,26}, and Na_3Bi ²⁷. This indicates that the linear MR can be caused by other mechanisms and takes place at rather low fields. For example, the mobility fluctuations due to disorder effects has been suggested to be the possible origin for the low field linear MR in Cd_3As_2 ².

In YbMnBi_2 , we have also observed linear MR in both in-plane (Fig. 1d in the revised manuscript) and interlayer (Fig. 2b) transport for $B \perp I$ when the magnetic field is above a few Tesla. More specifically, the $\rho_{zz}(B)$ data taken at $\theta = 90^\circ$ (Fig. 2b in the current manuscript) exhibits B^2 dependence in a low field range, but crossovers to a linear dependence above 10T. The interlayer transport for the field configuration of $\theta = 90^\circ$ should be dominated by the coherent momentum relaxation, since LL quantization is fully suppressed for this field orientation as mentioned above. According to this fact, we can assume similar field dependences of MR for the momentum relaxation channel for $\theta < 90^\circ$, i.e. the linear B -dependence for high fields, and B^2 -dependence for low fields. Thus the conductivity of the momentum relaxation channel can be assumed to $\sigma_c = \sigma_0 / (1 + k_1 \cdot B_{xy}^2)$ for low fields and $\sigma_0 / (1 + k_2 \cdot |B_{xy}|)$ for high fields. Although such an assumption is somewhat rough and does not reflect the crossover from the B - to B^2 -dependence in MR, the interlayer magnetoresistivity for $\theta < 90^\circ$ derived from our model $\rho_{zz}(B, \theta) = 1 / [\sigma_t^{LL0}(B, \theta) + \sigma_c(B, \theta)]$, which combines conduction contributions from both the tunneling and the momentum relaxation channels, fits to both the $\rho_{zz}(B)$ data under different θ (Fig. 2b) and the $\rho_{zz}(\theta)$ data (Fig. 2c) under different fields very well, indicating that our model has captured the main features of the interlayer transport.

We have added the above discussions to the revised manuscript. Again, we thanks the reviewer for seeing this problem.

2. According to Ref. 2, YbMnBi_2 appears to host several Weyl points not only at Fermi energy but also away from Fermi energy. It would be very helpful for the readers' understanding if the authors present the schematic band structure associated with the Weyl points.

This is a very good suggestion! Indeed, the first principle calculations suggest YbMnBi_2 has a number of Weyl points and the ARPES experiments probed the Weyl points at E_F ¹. Following the reviewer suggestion, we have added the schematics of the Fermi surface and band structure determined by the ARPES experiments¹ to Fig. 1.

3. The authors should distinguish the words ``Dirac' and ``Weyl'. I guess ``Weyl' fermion would be appropriate for YbMnBi_2 .

As we discussed above, both "Weyl" and "Dirac" bands exist in YbMnBi_2 . The electron- and hole-like pockets are comprised of the Dirac bands, while the Weyl points are formed by the Weyl band crossing. We have made this clear in the revised manuscript.

4. I would recommend the authors to refer to the studies reporting the interlayer resistivity for the

related Dirac materials, e.g., for SrMnBi₂ (Y. J. Jo, et al., PRL 113, 156602 (2014)) and EuMnBi₂ (H. Masuda et al., Sci. Adv. 2, e1501117 (2016)).

We thank the reviewer for pointing out this issue and apologize for not including these relevant references. In the revised manuscript, we have cited these two important works, as well as other relevant references.

Reference

1. Borisenko, S. *et al.* Time-Reversal Symmetry Breaking Type-II Weyl State in YbMnBi₂. *arXiv:1507.04847*, (2015).
2. Narayanan, A. *et al.* Linear Magnetoresistance Caused by Mobility Fluctuations in *n*-Doped Cd₃As₂. *Phys. Rev. Lett.* **114**, 117201, (2015).
3. Zhao, Y. *et al.* Anisotropic Fermi Surface and Quantum Limit Transport in High Mobility Three-Dimensional Dirac Semimetal Cd₃As₂. *Phys. Rev. X* **5**, 031037, (2015).
4. Zhang, C. *et al.* Quantum Phase Transitions in Weyl Semimetal Tantalum Monophosphide. *arXiv:1507.06301*, (2015).
5. Moll, P. J. W. *et al.* Magnetic torque anomaly in the quantum limit of Weyl semimetals. *Nature Communications* **7**, 12492, (2016).
6. Osada, T. Anomalous Interlayer Hall Effect in Multilayer Massless Dirac Fermion System at the Quantum Limit. *J. Phys. Soc. Jpn.* **80**, 033708, (2011).
7. Sato, M. *et al.* Transport Phenomenon of Multilayer Zero-Gap Conductor in the Quantum Limit. *J. Phys. Soc. Jpn.* **80**, 023706, (2011).
8. Wang, A. *et al.* Two-dimensional Dirac fermions in YbMnBi₂ antiferromagnet. *arXiv:1604.01009*, (2016).
9. Hu, J. *et al.* π Berry phase and Zeeman splitting of Weyl semimetal TaP. *Sci. Rep.* **6**, 18674, (2016).
10. Mao, Z. Q., Nelson, K. D., Jin, R., Liu, Y. & Maeno, Y. Observation of Andreev Surface Bound States in the 3-K Phase Region of Sr₂RuO₄. *Phys. Rev. Lett.* **87**, 037003, (2001).
11. Gutman, D. B. & Maslov, D. L. Anomalous *c*-Axis Transport in Layered Metals. *Phys. Rev. Lett.* **99**, 196602, (2007).
12. Cooper, S. L. & Gray, K. E. *Physical Properties of High Temperature Superconductors*. (World Scientific, 1994).
13. Wattamaniuk, W. J., Tidman, J. P. & Frindt, R. F. Tunneling Conductivity in 4Hb-TaS₂. *Phys. Rev. Lett.* **35**, 62-65, (1975).
14. Tyler, A. W., Mackenzie, A. P., NishiZaki, S. & Maeno, Y. High-temperature resistivity of Sr₂RuO₄: Bad metallic transport in a good metal. *Phys. Rev. B* **58**, R10107-R10110, (1998).
15. Analytis, J. G. *et al.* Effect of Irradiation-Induced Disorder on the Conductivity and Critical Temperature of the Organic Superconductor κ -(BEDT-TTF)₂Cu(SCN)₂. *Phys. Rev. Lett.* **96**, 177002, (2006).
16. Valla, T. *et al.* Coherence-incoherence and dimensional crossover in layered strongly correlated metals. *Nature* **417**, 627-630, (2002).
17. Osada, T. Negative Interlayer Magnetoresistance and Zero-Mode Landau Level in Multilayer Dirac Electron Systems. *J. Phys. Soc. Jpn.* **77**, 084711, (2008).

18. Ando, T. Physics of Graphene: Zero-Mode Anomalies and Roles of Symmetry. *Progress of Theoretical Physics Supplement* **176**, 203-226, (2008).
19. Pippard, A. B. *The Dynamics of Conduction Electrons*. (Gordon and Breach, 1965).
20. Pippard, A. B. *Magnetoresistance in Metals*. (Cambridge University Press, 1989).
21. Abrikosov, A. A. Quantum magnetoresistance. *Phys. Rev. B* **58**, 2788-2794, (1998).
22. Xu, R. *et al.* Large magnetoresistance in non-magnetic silver chalcogenides. *Nature* **390**, 57-60, (1997).
23. Myers, K. D. *et al.* Systematic study of anisotropic transport and magnetic properties of RAgSb₂ (R=Y, La–Nd, Sm, Gd–Tm). *J. Magn. Magn. Mater.* **205**, 27-52, (1999).
24. Wang, K. & Petrovic, C. Multiband effects and possible Dirac states in LaAgSb₂. *Phys. Rev. B* **86**, 155213, (2012).
25. He, L. P. *et al.* Quantum Transport Evidence for the Three-Dimensional Dirac Semimetal Phase in Cd₃As₂. *Phys. Rev. Lett.* **113**, 246402, (2014).
26. Liang, T. *et al.* Ultrahigh mobility and giant magnetoresistance in the Dirac semimetal Cd₃As₂. *Nature Mater.* **14**, 280-284, (2015).
27. Xiong, J. *et al.* Anomalous conductivity tensor in the Dirac semimetal Na₃Bi. *Europhys. Lett.* **114**, 27002, (2016).

Reviewers' comments:

Reviewer #1 (Remarks to the Author):

The authors have addressed all my concerns. The current manuscript is ready for the final publication in Nature Communications.

Reviewer #2 (Remarks to the Author):

In the revised manuscript, the authors give a clear picture of the observed unusual interlayer quantum transport, especially the sharp peak of AMR at large magnetic fields. The manuscript has been improved a lot. Almost all of the previous comments have been addressed properly. Before the recommendation of publication, I still have several questions.

1. The authors explain the unusual interlayer quantum transport behavior by considering both the 2D Weyl fermions and 2D electron- and hole-like Dirac fermions. The Fermi level is right at the Weyl node. The total interlayer magnetoresistivity in Eq. (2) are contributed by the zeroth Landau level of Weyl fermions and Dirac bands. They argue that in the quantum limit of the Weyl bands, the degeneracy of the zero LL increases and thus the density of state at the Fermi level increases, resulting in the unusual sharp peak in AMR. In the presence of the magnetic field, both the Weyl bands and Dirac bands are quantized to the Landau levels. The degeneracy of ANY Landau level (not only the zeroth LL) is $eB/2\pi$. They argued near "the quantum limit" the conductivity contributed by the zero LL of Weyl fermions exceeds those by the Dirac bands in Eq. (2), which results in the unusual behaviors. This is not very convincing. The density of states of any Landau level increases with increasing magnetic field, therefore the conductivity contribution σ_c should also increase. The authors should explain the reason more carefully why the zeroth LL of Weyl fermions plays a more important role at large magnetic fields.

2. The authors still do not distinguish Landau levels in 2D systems and Landau bands in 3D cases. In the first paragraph they stated that for both 2D and 3D topological materials, the quantized energies in the presence of magnetic field is $v_F \sqrt{2eBn}$. It is not true for 3D cases, where the Landau bands should also depend on the momentum that is parallel to the magnetic field, i.e., $v_F \sqrt{2eBn+k_z^2}$. Only for 2D topological materials with linear dispersion, the zero Landau level is always locked to the band crossing point.

Reviewer #3 (Remarks to the Author):

The authors significantly revised the manuscript following the reviewers' comments. The present manuscript clearly explains the electronic structures for YbMnBi₂, which has both Weyl-like and Dirac-like carriers. They also made clear the model of fitting for the AMR, based on the plausible band structures.

Having said that, still I do not think the main claim of this manuscript is experimentally supported at all. This is because the negative MR was not observed even when θ is close to zero. For θ close to zero, the contribution from the Dirac bands (i.e., σ_c) should be minimal because $B_{xy}=0$, whereas the contribution from the zeroth mode (i.e. $\sigma_t^{\{LL0\}}$) should be maximum, which naturally results in the negative MR if the tunneling of the zeroth modes plays a vital role. In experiments, however, small positive MR manifests itself even for $\theta=4$ degree, which verifies the contribution of tunneling between the zeroth modes to the interlayer conductivity is negligible if any. Therefore, I would strongly recommend the authors to remove their claims (in the main text and title) that the unusual AMR results from the zeroth LL mode and that the interlayer tunneling of the zeroth modes is remarkable, before the manuscript is published.

What follow are minor comments.

- (1) The authors assume that σ_c is expressed as $\sigma_0/(1+k_1 B_{xy}^2)$, which may be experimentally checked for $\theta=90$ degree, by using the data of R_{zz} and R_{zx} .
- (2) In Fig. 2c, the authors mention that they observed the $\sin^2(\theta)$ dependence of the AMR at low field (0.5 T), which is quite hard to see in the present scale.
- (3) The authors should clarify the value of field angle where they estimate the values of w_1 and w_2 in Fig. 3d.

Response to Reviewer #1's comment:

The authors have addressed all my concerns. The current manuscript is ready for the final publication in Nature Communications.

We thank Reviewer #1 for his/her recommendation for publication of our manuscript in Nature Communications.

Response to Reviewer #2's comments:

In the revised manuscript, the authors give a clear picture of the observed unusual interlayer quantum transport, especially the sharp peak of AMR at large magnetic fields. The manuscript has been improved a lot. Almost all of the previous comments have been addressed properly. Before the recommendation of publication, I still have several questions.

First, we thank the Reviewer #2 for taking time to review our revised manuscript. We really appreciate his/her insightful comments which have been very helpful in further improving our manuscript. We have made point-to-point response (in black) to the reviewer's questions (in red).

1. The authors explain the unusual interlayer quantum transport behavior by considering both the 2D Weyl fermions and 2D electron- and hole-like Dirac fermions. The Fermi level is right at the Weyl node. The total interlayer magnetoresistivity in Eq. (2) are contributed by the zeroth Landau level of Weyl fermions and Dirac bands. They argue that in the quantum limit of the Weyl bands, the degeneracy of the zero LL increases and thus the density of state at the Fermi level increases, resulting in the unusual sharp peak in AMR. In the presence of the magnetic field, both the Weyl bands and Dirac bands are quantized to the Landau levels. The degeneracy of ANY Landau level (not only the zeroth LL) is $eB/2\pi$. They argued near "the quantum limit" the conductivity contributed by the zero LL of Weyl fermions exceeds those by the Dirac bands in Eq. (2), which results in the unusual behaviors. This is not very convincing. The density of states of any Landau level increases with increasing magnetic field, therefore the conductivity contribution σ_c should also increase. The authors should explain the reason more carefully why the zeroth LL of Weyl fermions plays a more important role at large magnetic fields.

The reviewer asked a very good question. We apologize for not addressing this clearly in the previous version of manuscript. Although the LL degeneracy of Dirac bands is also enhanced upon increasing field, it should not contribute to the unusual interlayer transport properties probed in AMR (Fig. 2c) and Hall resistance (Fig.3d) at high fields. Since the quantum oscillation frequencies of Dirac fermions are high (115T and 162T, see Fig. 1d), the quantum limit of the Dirac bands cannot be reached until the field is increased above 230T. Given that our experiments were conducted below 31T, the variation of LL

degeneracy should be small for the Dirac bands. Therefore, the variation of the DOS(E_F) of the Dirac bands with the field rotation in the field range of our experiments is expected to be small, inconsistent with our experimental observation of the drastic changes of AMR above 9T and Hall resistance above 6T. The AMR of the Dirac fermion transport channel should more or less follow the classical Lorentz effect, *i.e.* $AMR(\theta) \propto B_{xy}^2 = B^2 \sin^2 \theta$, which is indeed observed at low fields but not consistent with the unusual features at high fields (see Fig. 2c in manuscript).

In contrast, the quantum limit can be easily achieved for the Weyl bands in YbMnBi₂, since its Weyl nodes are at the Fermi level [1]. Its zeroth LL degeneracy would increase drastically near the quantum limit, which should result in a significant increase of DOS(E_F) at the zeroth LL and be responsible for the unusual AMR and angular dependence of Hall resistance. This interpretation is supported by our successful fits of the $\rho_{zz}(B)$, $AMR(\theta)$ and $R_{xx}(\theta)$ data to a model which considers the interlayer quantum tunneling transport of the zeroth LL's Weyl fermions.

We have added some of the above discussions to the revised manuscript to address the reviewer's question (highlighted on Page 14).

2. The authors still do not distinguish Landau levels in 2D systems and Landau bands in 3D cases. In the first paragraph they stated that for both 2D and 3D topological materials, the quantized energies in the presence of magnetic field is $v_F \sqrt{2e\hbar B |n|}$. It is not true for 3D cases, where the Landau bands should also depend on the momentum that is parallel to the magnetic field, *i.e.*, $v_F \sqrt{2e\hbar B |n| + k_z^2}$. Only for 2D topological materials with linear dispersion, the zero Landau level is always locked to the band crossing point.

We really appreciate that the Reviewer found this important problem. We apologize for overlooking it. Indeed, as pointed out by the reviewer, the Landau level energy for 2D Dirac/Weyl fermions is $\varepsilon_n = \pm v_F \sqrt{2e\hbar B |n|}$ ($n=0, \pm 1, \pm 2 \dots$) [2]. However, for a 3D Dirac/Weyl system, the Landau level energy depends on the momentum along the field direction (denoted as k_z), and $\varepsilon_n = \pm v_F \sqrt{2e\hbar B |n| + k_z^2}$ ($n=0, \pm 1, \pm 2 \dots$) [3]. Therefore, the zeroth Landau level is not locked to the Dirac/Weyl node for a 3D Dirac/Weyl system.

For YbMnBi₂, previous studies have shown that its Weyl bands are of strong 2D character [1]. The Fermi velocity of the Weyl fermions along the k_z direction is found to be more than two orders of magnitude smaller than that along k_x . Therefore, we can approximately treat the Landau levels of the Weyl bands in YbMnBi₂ as LLs in 2D systems. In the current revised manuscript, we have made revisions to both the introduction and discussion sections to address this issue (the revised sections have been highlighted on Pages 2 and 3, as well as the figure caption to Fig. 1).

Response to Reviewer #3's comments:

The authors significantly revised the manuscript following the reviewers' comments. The present manuscript clearly explains the electronic structures for YbMnBi₂, which has both Weyl-like and Dirac-like carriers. They also made clear the model of fitting for the AMR, based on the plausible band structures.

First, we thank the Reviewer #3 for taking time to review our revised manuscript. We appreciate his/her comments which have been very helpful in further improving our manuscript. We have made point-to-point response (in black) to the reviewer's questions (in red).

Having said that, still I do not think the main claim of this manuscript is experimentally supported at all. This is because the negative MR was not observed even when θ is close to zero. For θ close to zero, the contribution from the Dirac bands (i.e., σ_c) should be minimal because $B_{xy}=0$, whereas the contribution from the zeroth mode (i.e., σ_t^{LL0}) should be maximum, which naturally results in the negative MR if the tunneling of the zeroth modes plays a vital role. In experiments, however, small positive MR manifests itself even for $\theta=4^\circ$, which verifies the contribution of tunneling between the zeroth modes to the interlayer conductivity is negligible if any. Therefore, I would strongly recommend the authors to remove their claims (in the main text and title) that the unusual AMR results from the zeroth LL mode and that the interlayer tunneling of the zeroth modes is remarkable, before the manuscript is published.

We thank Reviewer#3 for bringing up this issue. Indeed, the tunneling conductance should be enhanced upon increasing the perpendicular field component, and results in negative MR. Reviewer#3 asked the same question in the first round of review. In our previously submitted response letter, we have provided the interpretation for the absence of negative MR: the positive MR is due to the coexistence of the tunneling and the momentum relaxation channels in YbMnBi₂. The weak positive MR for $\theta=4^\circ$ (B nearly parallel to I) is due to the competition between the negative MR component from the tunneling channel and the positive MR component from the momentum relaxation channel. However, we did not provide justification for the positive longitudinal (i.e. $B//I$) MR for the momentum relaxation channel in our previous response, which might cause Reviewer#3 to be confused. We apologize for missing that point and give more detailed explanations below.

The classical theory based on a free electron model yields a quadratic field dependence for MR, due to the elongated electron trajectories caused by Lorentz effect. Therefore, zero longitudinal MR is expected due to zero Lorentz force. However, the longitudinal MR is usually finite and could have relatively large values in real materials, such as the isostructure compounds AMn(Bi/Sb)₂ (A=Sr, Ba, Ca) [4-9]. Such positive longitudinal MR for $B//I$ can be understood by taking the following factor into account. According to a recent theoretical study by Pal and Maslov [10], anisotropic Fermi surface can cause striking positive longitudinal MR, ranging from a few to a few tens of a percent. Using the

semiclassical Boltzmann equation, Pal and Maslov showed electrons moving with the current fluctuate along the transverse direction, thus experiencing non-zero Lorentz force when the Fermi surface is anisotropic. For YbMnBi₂, its Fermi surface is indeed strongly anisotropic as revealed by ARPES studies [1]. Thus, we can reasonably expect a positive longitudinal MR for YbMnBi₂.

With this consideration, we can better understand the weak positive MR for $\theta=4^\circ$ in YbMnBi₂. Though the tunneling channel is expected to have negative MR, its competition with the positive MR of the momentum relaxation channel leads to a very small positive MR for $\theta=4^\circ$.

The interlayer quantum tunneling of the zeroth LL mode have been demonstrated by both the interlayer AMR and the Hall effect in our experiments. In both measurements, we have observed the crossover from the low field classical behavior to the high field unusual angular dependence, which can be well described by a model which considers the coexistence of the tunneling process of the zeroth LL mode of Weyl fermions and the momentum relaxation process of Dirac fermions. As far as we know, no other mechanism can give satisfactory interpretations for these observations.

What follow are minor comments.

We thank the Reviewer#3 for raising these issues.

(1) The authors assume that σ_c is expressed as $\sigma_0/(1+k_1 B_{xy}^2)$, which may be experimentally checked for $\theta=90^\circ$, by using the data of R_{zz} and R_{zx} .

We have indeed shown the ρ_{zz} data at $\theta=90^\circ$ ($B \perp I$) in Fig. 2b. As stated in the manuscript, we observed B^2 dependence in a low field region, which gradually evolves to a linear field dependence above 10T. As we have mentioned in the previous response letter, though linear MR is expected to occur at quantum limit theoretically, linear MR has also been widely observed in many materials at relatively low fields when quantum limit is not reached. One possible interpretation is the mobility fluctuations due to disorder effects.

As for R_{zx} , since it is a transverse (Hall) resistance, it does not follow the B^2 dependence expected for the classical *longitudinal* magnetoresistance.

(2) In Fig. 2c, the authors mention that they observed the $\sin^2(\theta)$ dependence of the AMR at low field (0.5 T), which is quite hard to see in the present scale.

We have included a zoomed-in data in the inset of Fig. 2c, showing the $\sin^2\theta$ -dependence for the AMR at $B=0.1$ T

(3) The authors should clarify the value of field angle where they estimate the values of w_1 and w_2 in Fig. 3d.

We preformed the fits of the Hall resistance in the whole angular range (0° - 180°).

- [1] S. Borisenko, D. Evtushinsky, Q. Gibson, A. Yaresko, *et al.*, *arXiv:1507.04847* (2015)
- [2] T. Ando, *Progress of Theoretical Physics Supplement* **176**, 203-226 (2008)
- [3] B. A. Bernevig and L. H. Taylor, *Topological Insulators and Topological Superconductors*. Princeton University Press: 2013.
- [4] K. Wang, D. Graf, H. Lei, S. W. Tozer, *et al.*, *Phys. Rev. B* **84**, 220401 (2011)
- [5] L. Li, K. Wang, D. Graf, L. Wang, *et al.*, *Phys. Rev. B* **93**, 115141 (2016)
- [6] K. Wang, D. Graf, L. Wang, H. Lei, *et al.*, *Phys. Rev. B* **85**, 041101 (2012)
- [7] K. Wang, D. Graf and C. Petrovic, *Phys. Rev. B* **87**, 235101 (2013)
- [8] J. Liu, J. Hu, H. Cao, Y. Zhu, *et al.*, *Sci. Rep.* **6**, 30525 (2016)
- [9] J.Y. Liu, J. Hu, D. Graf, S. M. A. Radmanesh, *et al.*, *arXiv:1507.07978* (2015)
- [10] H. K. Pal and D. L. Maslov, *Phys. Rev. B* **81**, 214438 (2010)

Reviewers' comments:

Reviewer #2 (Remarks to the Author):

In this revised version, the manuscript has been improved a lot. However, there is a minor comment should be addressed properly before the final publication on Nature Communications.

The authors use Eq. (5) to analyze the angular dependence of the Hall resistance. They claim that this equation reproduces all the angular dependence for different magnitudes of the magnetic field. However, this expression may not be proper. When both Weyl and Dirac bands are taken into account, the total conductivity is a weighted summation of the conductivities from Weyl and Dirac bands. Then they obtained the total Hall resistivity by directly inverting the Hall conductivity. This process is not proper because the resistivity should also depend on the longitudinal conductivity. The authors should explain or at least comment on why the longitudinal conductivity σ_{zz} does not affect the Hall resistivity.

Reviewer #3 (Remarks to the Author):

The authors have explained the possibility of the positive MR for $\theta=0^\circ$ (longitudinal magnetoresistance) that competes with the negative MR caused by the zero mode tunneling. I agree that the positive longitudinal MR may exist in the present compound, as predicted in ref. 34. However, its magnitude seems to be typically the order of 10 percent at the largest. The fact that the observed MR remains positive even near $\theta=0^\circ$ again means that the negative MR from the zero mode, if any, contributes to the interlayer transport much less than the small positive longitudinal MR from the Dirac-like carriers. This is quite different from the case of the layered organic conductor, where the negative MR easily exceeds the small positive MR above 0.2 T, resulting in a large decrease in resistivity (in ref. 33, e.g. the resistivity at 3 T is approximately 1/3 of that at 0.2 T). Therefore, based on the data of the longitudinal MR ($\theta=4^\circ$), the tunneling process of the zero mode should play the minimal role on the interlayer conduction in YbMnBi₂, as I repeated in the previous comments. Nevertheless, the AMR and interlayer Hall effects might suggest the possible contribution of the zero mode tunneling. In particular, as shown in the inset to Fig 3, the value of w_1/w_2 is close to 1 for $B=9$ T, which means the zero mode tunneling process contributes to the interlayer Hall resistivity to a similar extent to the Dirac bands' momentum relaxation. If this was true, the negative MR would manifest itself near $\theta=0^\circ$ at high fields. I would recommend the authors to *quantitatively* discuss this apparent inconsistency in the main text. (The phrases “considerable positive longitudinal MR” in the 269th and 273rd lines are misleading.)

I am also confused about the fitted results in Fig. 2b. The authors seems to fit the data for $\theta=4^\circ$ by assuming $\sigma_c = \sigma_0/(1+k_1 B_{xy}^2)$ or $\sigma_0/(1+k_2 B_{xy})$. However, this is inconsistent with the authors' new claim that the positive longitudinal MR (expressed by B_z not B_{xy}) may matter for $\theta\sim 0^\circ$, as discussed in the paragraph starting from 263rd line. The authors should explain the details of fitting of the data for $\theta=4^\circ$ by considering the longitudinal MR.

In the comment (1) in my previous review, what I meant was that the field dependence of σ_c can be experimentally obtained for $\theta=90^\circ$. In fact, σ_{zz} is expressed for $\theta=90^\circ$ (when $\rho_{yx}=0$) as

$$\sigma_{zz} = \frac{\rho_{xx}}{\rho_{xx}\rho_{zz} + \rho_{xz}^2}$$

Because $\sigma^{LL0} \sim 0$ for $\theta=90^\circ$, the total σ_{zz} should be equal to σ_c , which can be calculated by the experimental values of ρ_{xx} , ρ_{xz} , and ρ_{zz} . The authors should show the obtained σ_{zz} as a function of field, so they can check if the assumption that $\sigma_c = \sigma_0/(1+k_1 B_{xy}^2)$ or $\sigma_0/(1+k_2 B_{xy})$ is valid. With respect to this point, the authors assume that $\sigma_{zz}=1/\rho_{zz}$ and $\sigma_{zx}=1/\rho_{xz}$.

in eq. (2) and (5), which is not correct in general. For instance, from a back-of-the-envelope calculation, σ_{zx} is written by (please see also ref. 41)

$$\sigma_{zx} = \frac{\rho_{xx}\rho_{xz}}{\rho_{xx}^2\rho_{zz} + \rho_{xy}^2\rho_{zz} + \rho_{xz}^2\rho_{xx}}$$

The authors should clearly explain why Eq. (2) and (5) works fine for the present case. If not, I would recommend the authors to recalculate them by using more precise equations.

After revising the above points, I would recommend the publication of this manuscript.

Response to Reviewer #2's comments:

In this revised version, the manuscript has been improved a lot. However, there is a minor comment should be addressed properly before the final publication on Nature Communications.

We thank Reviewer #2 for taking the time to review our revised manuscript and for his/her positive comments. We also appreciate the additional minor comment which has been very helpful in further improving our manuscript.

The authors use Eq. (5) to analyze the angular dependence of the Hall resistance. They claim that this equation reproduces all the angular dependence for different magnitudes of the magnetic field. However, this expression may not be proper. When both Weyl and Dirac bands are taken into account, the total conductivity is a weighted summation of the conductivities from Weyl and Dirac bands. Then they obtained the total Hall resistivity by directly inverting the Hall conductivity. This process is not proper because the resistivity should also depend on the longitudinal conductivity. The authors should explain or at least comment on why the longitudinal conductivity σ_{zz} does not affect the Hall resistivity.

We extremely appreciate that Reviewer 2 pointed out this important issue which was overlooked in our previous manuscript. In general, the conversion between resistivity and conductivity requires tensor transformation. As the reviewer pointed out, it is not appropriate to obtain Hall resistivity by directly inverting Hall conductivity. To ensure the validity of our analyses, we have performed additional measurements on $\rho_{xx}(\theta)$, $\rho_{yy}(\theta)$, $\rho_{yx}(\theta)$ and $\rho_{xy}(\theta)$ [see Figs. 1a-1d for Reviewer 2]. These newly measured data, together with previously measured $\rho_{zz}(\theta)$ and $\rho_{zx}(\theta)$ [Figs. 1e and 1f for Reviewer 2], enable us to create the resistivity tensor and obtain the Hall conductivity σ_{xz} through tensor inversion. Based on the double transport channel model discussed in the manuscript, the total Hall conductivity in YbMnBi₂ can be expressed as:

$$\sigma_{zx}^{total} = w_1 \cdot \sigma_{zx}^{LL0} + w_2 \cdot \sigma_{zx}^C \quad (1)$$

where σ_{zx}^{LL0} and σ_{zx}^C represent the Hall conductivities contributed by Weyl and Dirac bands, respectively.

w_1 and w_2 represent the weights of the contribution for each type of band. σ_{zx}^C due to the momentum relaxation channel of the Dirac bands can be found from the Boltzmann transport theory, i.e.

$$\sigma_{zx}^C = \sigma_0 \frac{\omega_c \tau}{1 + (\omega_c \tau)^2}, \quad (2)$$

where $\sigma_0 = \frac{ne^2\tau}{m^*}$ and $\omega_c = \frac{eB \sin \theta}{m^*}$.

For the tunneling channel of the zeroth LLs, its Hall conductivity σ_{zx}^{LL0} under a field oriented at an angle of θ (inset to Fig. 1c for Reviewer 2) can be expressed as the following equation according to the theoretical study by Osada [1],

$$\sigma_{zx}^{LL0}(B, \theta) = a \frac{B_y}{B_z^2} \exp\left(-b \frac{B_y^2}{B_z}\right) \quad (3),$$

where a and b are material dependent constants, $B_y = B \sin \theta$ and $B_z = B \cos \theta$.

The total Hall conductivity σ_{zx}^{total} can be obtained via taking the inverse of the resistivity tensor. In a 3D material, the resistivity ($\hat{\rho}$) and conductivity ($\hat{\sigma}$) tensors can be expressed as:

$$\hat{\rho} = \begin{bmatrix} \rho_{xx} & \rho_{xy} & \rho_{xz} \\ \rho_{yx} & \rho_{yy} & \rho_{yz} \\ \rho_{zx} & \rho_{zy} & \rho_{zz} \end{bmatrix}, \quad \hat{\sigma} = \begin{bmatrix} \sigma_{xx} & \sigma_{xy} & \sigma_{xz} \\ \sigma_{yx} & \sigma_{yy} & \sigma_{yz} \\ \sigma_{zx} & \sigma_{zy} & \sigma_{zz} \end{bmatrix}, \quad \text{and} \quad \hat{\sigma} = \hat{\rho}^{-1}. \quad (4)$$

The resistivity tensor elements ρ_{ij} ($i, j = x, y, z$) can be directly obtained by measuring the voltage along the $+j$ direction with the current flowing along the $+i$ direction. For the experimental setup shown in the insets to Fig. 1a-1f (for Reviewer 2) where $\mathbf{B}=(0, B\sin\theta, B\cos\theta)$, the resistivity tensor elements ρ_{yz} , ρ_{zy} , ρ_{xy} , ρ_{yx} , ρ_{xz} and ρ_{zx} are expected to have the following relations: $\rho_{yz} = \rho_{zy} = 0$, $\rho_{xy} = -\rho_{yx}$ and $\rho_{xz} = -\rho_{zx}$. The first relationship of $\rho_{yz} = \rho_{zy} = 0$ is obvious and need not be verified. The latter two relations were verified with additional measurements, as shown in Figs. 1c-1d and the right inset to Fig. 1e. By taking the inverse the resistivity tensor, the diagonal (Hall) conductivity tensor element σ_{zx} can be written as:

$$\sigma_{zx} = \frac{\rho_{yy}\rho_{xz}}{\rho_{xx}\rho_{yy}\rho_{zz} - \rho_{xy}\rho_{yx}\rho_{zz} - \rho_{xz}\rho_{zx}\rho_{yy}} \quad (5).$$

Fig. 1 (for Reviewer 2) (a-f) Angular dependences of resistivity tensor elements; (a) ρ_{xx} , (b) ρ_{yy} , (c) ρ_{xy} , (d) ρ_{yx} , (e) ρ_{zx} , and (f) ρ_{zz} at different fields from 0.5 to 9T. The inset in each panel shows the experimental setup. The right inset in (e) shows $\rho_{xz} \approx -\rho_{zx}$ at $B=9\text{T}$. (g) Angular dependence of Hall conductivity $\sigma_{zx}(\theta)$. The solid lines in (g) represent fits to Eq. (1). The data have been shifted for clarity.

Using the measured resistivity tensor elements shown in Fig. 1a-1f and Eq. (5), we derived the angular dependence of the total Hall conductivity $\sigma_{zx}^{total}(\theta)$ as shown in Fig. 1g. $\sigma_{zx}^{total}(\theta)$ displays a $\sin\theta$ -like dependence at lower fields, but strongly deviates from it at high fields; a local minimum at $\theta=90^\circ$ gradually develops when the field is increased above 1 T. Such an unusual evolution of $\sigma_{zx}^{total}(\theta)$ with magnetic field cannot be described by the classical transport model (Eq. 2), but can be understood by considering the interlayer zeroth LL quantum tunneling: At low fields, when the zeroth and 1st LLs are

not well separated, the quantum tunneling is minimized so that the total Hall conductivity is dominated by the classical momentum relaxation transport of Dirac bands (Eq. 2). Thus, it exhibits a $\sin\theta$ dependence as predicted by Eq. (2) which can be approximated to $\sigma_{zx}^C \propto \omega\tau \propto B \sin\theta$ for low fields. However, at high fields, the interlayer zeroth LL tunneling becomes important, thus leading to unusual angular dependence of σ_{zx}^{total} . The local minimum of σ_{zx}^{total} at $\theta=90^\circ$ is caused by the suppression of the 2D Landau level quantization when the field is oriented close to the in-plane direction ($\theta=90^\circ$). This interpretation is verified by the quantitative fit of σ_{zx}^{total} to Eq. 1 (Fig. 1g). Furthermore, as discussed in the manuscript, the $\tan\theta$ dependence of ρ_{zx} in the lower angle range (Fig. 3d in the current revised manuscript) as well as the unusual angular dependence of $\rho_{zz}(\theta)$ at high fields (Fig. 2c in the manuscript) also provide strong support for the interlayer zeroth LL tunneling.

Although the Hall conductivity σ_{zx} cannot be directly taken as the inverse of Hall resistivity ρ_{zx} as discussed above, the longitudinal conductivity σ_{zz} can be approximated to $1/\rho_{zz}$ as discussed below. In general, σ_{zz} should be obtained also through tensor conversion (see Supplementary Note 2), *i.e.*

$$\sigma_{zz} = \frac{\rho_{xx}\rho_{yy} - \rho_{xy}\rho_{yx}}{\rho_{xx}\rho_{yy}\rho_{zz} - \rho_{xy}\rho_{yx}\rho_{zz} - \rho_{xz}\rho_{zx}\rho_{yy}} \quad (6)$$

As shown in Figs. 1a-1f (for Reviewer 2), ρ_{xy} and ρ_{yx} are 2-3 orders of magnitude smaller than all other tensor elements, and $\rho_{xx}\rho_{yy}\rho_{zz} \gg \rho_{xz}\rho_{zx}\rho_{yy}$. Eq. 6 is approximated to $\sigma_{zz} \approx 1/\rho_{zz}$ when all of its trivial terms, including $\rho_{xy}\rho_{yx}$, $\rho_{xy}\rho_{yx}\rho_{zz}$, and $\rho_{xz}\rho_{zx}\rho_{yy}$, are ignored. To further demonstrate its validity, we made a direct comparison between the inverse of ρ_{zz} and the σ_{zz} calculated using resistivity tensor elements and found they are almost the same, as shown Fig. 2c (for Reviewer 2 and 3, see Page 6). Therefore, our fits for $\rho_{zz}(B, \theta)$ in our previous manuscript (Fig. 2b and 2c), which is based on the assumption of $\sigma_{zz} \approx 1/\rho_{zz}$, is correct.

Again, we thank the referee for pointing out this important issue. We have added some of the above discussions and new data to the revised manuscript and the detailed resistivity-conductivity tensor transformation analyses to the Supplementary Information.

Response to Reviewer #3's comments:

We thank Reviewer #3 for carefully reading our revised manuscript. We also appreciate his/her comments which have been very helpful in further improving our manuscript.

The authors have explained the possibility of the positive MR for $\theta=0^\circ$ (longitudinal magnetoresistance) that competes with the negative MR caused by the zero mode tunneling. I agree that the positive longitudinal MR may exist in the present compound, as predicted in ref. 34. However, its magnitude seems to be typically the order of 10 percent at the largest. The fact that the observed MR remains positive even near $\theta=0^\circ$ again means that the negative MR from the zero mode, if any, contributes to the interlayer transport much less than the small positive longitudinal MR from the Dirac-like carriers. This is quite different from the case of the layered organic conductor, where the negative MR easily exceeds the small positive MR above 0.2 T, resulting in a large decrease in resistivity (in ref. 33, e.g. the resistivity at 3 T is approximately 1/3 of that at 0.2 T). Therefore, based on the data of the longitudinal MR ($\theta=4^\circ$), the tunneling process of the zero mode should play the minimal role on the interlayer conduction in YbMnBi₂, as I repeated in the previous comments. Nevertheless, the AMR and interlayer

Hall effects might suggest the possible contribution of the zero mode tunneling. In particular, as shown in the inset to Fig 3, the value of w_1/w_2 is close to 1 for $B=9$ T, which means the zero mode tunneling process contributes to the interlayer Hall resistivity to a similar extent to the Dirac bands' momentum relaxation. If this was true, the negative MR would manifest itself near $\theta = 0^\circ$ at high fields. I would recommend the authors to quantitatively discuss this apparent inconsistency in the main text. (The phrases "considerable positive longitudinal MR" in the 269th and 273rd lines are misleading.)

We appreciate that the referee brought up this question again. Motivated by the referee's comments, we have now found a more reasonable answer to the question of why the interlayer longitudinal magnetoresistance (LMR) remains positive with a small magnitude, even though the interlayer tunneling of the zeroth LL's fermions plays an important role. In our last submission, we attributed the observed small positive LMR to the competition between the negative MR component due to the zeroth LL's tunneling channel of the Weyl bands and the positive MR component due to the momentum relaxation channel of the Dirac bands. We provided one possible interpretation for the positive LMR component, *i.e.*, the anisotropic Fermi surface. As we pointed out in the manuscript, anisotropic Fermi surface is generally expected to result in only a few tens of percent positive LMR at maximum [3]. If the positive LMR component in YbMnBi_2 is driven purely by its anisotropic Fermi surface, its magnitude of negative LMR component attributed to the tunneling channel would also be around an order of a few tens of percent. As indicated by the referee, this seems unreasonably small when the tunneling channel plays a dominant role. We have carefully considered this question again and found we had overlooked an important factor which could cause very large positive LMR component in YbMnBi_2 , as discussed below.

In fact, large positive LMR is a generic feature of topological semimetals. For example, $\text{AMn}(\text{Bi/Sb})_2$ ($A=\text{Sr, Ba, Ca}$), which are isostructural to YbMnBi_2 and Dirac materials, have remarkable positive LMR for interlayer transport. LMR reaches 200% for BaMnBi_2 [4], 500% for BaMnSb_2 [5] and 300% for SrMnSb_2 [6] at 9T and $\sim 2\text{K}$, even as high as 10,000% at 31T for SrMnSb_2 [6]. Such large positive LMR can be attributed to their Dirac band transport. Since the Dirac nodes in these materials are far away from the Fermi level, their interlayer transport should not involve the zeroth LL's tunneling. Although the mechanism of the large LMR in Dirac materials has not been well understood, we can reasonably expect a very large positive LMR component resulting from the Dirac band transport channel in YbMnBi_2 due to its structural similarity to $\text{AMn}(\text{Bi/Sb})_2$. However, our observed LMR for ρ_{zz} in YbMnBi_2 reaches only 7% at $B=9\text{T}$ and 20% at $B=31\text{T}$ (Fig. 2b in the manuscript), which are one or two orders of magnitude smaller than those of $\text{AMn}(\text{Bi/Sb})_2$ materials. The strong suppression of positive LMR in YbMnBi_2 implies that its large positive LMR component expected for the Dirac band transport channel must be canceled by a large negative MR component caused by the zeroth LL tunneling channel of the Weyl bands. Given that the positive LMR component for ρ_{zz} in $\text{AMn}(\text{Bi/Sb})_2$ is within the 200%-10000% range, we anticipate the negative LMR in YbMnBi_2 has a magnitude within a similar range. As discussed in the revised manuscript, this interpretation is well supported by the observation of unusual angular dependences of ρ_{zz} and ρ_{zx} as well as the fits of ρ_{zz} and σ_{zx} to the double channel transport model which considers both interlayer Dirac band transport and zeroth LL quantum tunneling of the Weyl bands.

Again, we thank Reviewer#3 for raising this question. We have made revisions to the LMR discussions in the revised manuscript.

In the comment (1) in my previous review, what I meant was that the field dependence of σ_c can be experimentally obtained for $\theta=90^\circ$. In fact, σ_{zz} is expressed for $\theta=90^\circ$ (when $\sigma_{yx}=0$) as

$$\sigma_{zz} = \frac{\rho_{xx}}{\rho_{xx}\rho_{zz} + \rho_{xz}^2}$$

Because $\sigma^{LL0} \sim 0$ for $\theta=90^\circ$, the total σ_{zz} should be equal to σ_c , which can be calculated by the experimental values of ρ_{xx} , ρ_{xz} , and ρ_{zz} . The authors should show the obtained σ_{zz} as a function of field, so they can check if the assumption that $\sigma_c = \sigma_0/(1+k_1B_{xy}^2)$ or $\sigma_0/(1+k_2B_{xy})$ is valid. With respect to this point, the authors assume that $\sigma_{zz}=1/\rho_{zz}$ and $\sigma_{zx}=1/\rho_{zx}$ in eq. (2) and (5), which is not correct in general. For instance, from a back-of-the-envelope calculation, σ_{zx} is written by (please see also ref. 41)

$$\sigma_{zx} = \frac{\rho_{xx}\rho_{xz}}{\rho_{xx}^2\rho_{zz} + \rho_{xy}^2\rho_{zz} + \rho_{xz}^2\rho_{xx}}$$

The authors should clearly explain why Eq. (2) and (5) works fine for the present case. If not, I would recommend the authors to recalculate them by using more precise equations.

We extremely appreciate that Reviewer 3 pointed out this important issue which was overlooked in our previous manuscript. We followed Referee 3's suggestion and performed additional measurements (see Fig. 1a-1f for Reviewer 3) to create the resistivity tensor, and obtained the conductivity elements σ_{zz} and σ_{zx} through tensor inversion.

Fig. 1 (for Reviewer 3) (a-f) Angular dependences of resistivity tensor elements; (a) ρ_{xx} , (b) ρ_{yy} , (c) ρ_{xy} , (d) ρ_{yx} , (e) ρ_{zx} , and (f) ρ_{zz} at different fields from 0.5 to 9T and 2K. The inset in each panel shows the experimental setup. The right inset in (e) shows $\rho_{xz} \approx -\rho_{zx}$ at $B=9T$ and 2K. (g) Angular dependence of Hall conductivity $\sigma_{zx}(\theta)$. The solid lines in (g) represent fits to Eq. (11). The data have been shifted for clarity.

In a 3D material, the resistivity ($\hat{\rho}$) and conductivity ($\hat{\sigma}$) tensors can be expressed as:

$$\hat{\rho} = \begin{bmatrix} \rho_{xx} & \rho_{xy} & \rho_{xz} \\ \rho_{yx} & \rho_{yy} & \rho_{yz} \\ \rho_{zx} & \rho_{zy} & \rho_{zz} \end{bmatrix}, \quad \hat{\sigma} = \begin{bmatrix} \sigma_{xx} & \sigma_{xy} & \sigma_{xz} \\ \sigma_{yx} & \sigma_{yy} & \sigma_{yz} \\ \sigma_{zx} & \sigma_{zy} & \sigma_{zz} \end{bmatrix}, \quad \text{and } \hat{\sigma} = \hat{\rho}^{-1}. \quad (7)$$

The resistivity tensor elements ρ_{ij} ($i, j = x, y, z$) can be directly obtained by measuring the voltage along the $+j$ direction with current flowing along the $+i$ direction. For the experimental setup shown in the insets to Fig. 1a-1f (for Reviewer 3) where $\mathbf{B}=(0, B\sin\theta, B\cos\theta)$, the resistivity tensor elements ρ_{yz} , ρ_{zy} , ρ_{xy} , ρ_{yx} , ρ_{xz} and ρ_{zx} are expected to have the following relations: $\rho_{yz} = \rho_{zy} = 0$, $\rho_{xy} = -\rho_{yx}$ and $\rho_{xz} = -\rho_{zx}$. The first relationship of $\rho_{yz} = \rho_{zy} = 0$ is obvious and need not be verified. The latter two relations were verified with additional measurements, as shown in Figs. 1c-1d and the right inset to Fig. 1e (for Reviewer 3). By taking the inverse of the resistivity tensor, the conductivity tensor elements σ_{zz} and σ_{zx} can thus be derived as follows (the details are shown in Supplemental Note 2):

$$\sigma_{zz} = \frac{\rho_{xx}\rho_{yy} - \rho_{xy}\rho_{yx}}{\rho_{xx}\rho_{yy}\rho_{zz} - \rho_{xy}\rho_{yx}\rho_{zz} - \rho_{xz}\rho_{zx}\rho_{yy}} \quad (8),$$

$$\sigma_{zx} = \frac{\rho_{yy}\rho_{xz}}{\rho_{xx}\rho_{yy}\rho_{zz} - \rho_{xy}\rho_{yx}\rho_{zz} - \rho_{xz}\rho_{zx}\rho_{yy}} \quad (9).$$

Eq. (8) is reduced to the σ_{zz} equation given by Reviewer 3 when $\theta = 90^\circ$. Eq. (9) is identical to the σ_{zx} equation given by Reviewer 3 when we consider $\rho_{xx} = \rho_{yy}$, $\rho_{xy} = -\rho_{yx}$, $\rho_{xz} = -\rho_{zx}$. In our experimental setup, the relation of $\rho_{xy} = -\rho_{yx}$ and $\rho_{xz} = -\rho_{zx}$ are valid as demonstrated by the data shown in Fig. 1c and 1d and the inset to Fig. 1e; but the relation of $\rho_{xx} = \rho_{yy}$ does not hold, as shown by the data in Fig. 1a and 1b. Again, we appreciate Reviewer 3's guide for this data analysis.

In our previous manuscript, we expressed the overall interlayer magnetoresistivity as

$$\rho_z(B, \theta) = 1/[\sigma_t^{LL0}(B, \theta) + \sigma_c(B, \theta)] \quad (10),$$

which is based on the assumption that $\sigma_{zz} \approx 1/\rho_{zz}$. This assumption is valid as demonstrated below.

For the in-plane magnetic field ($\theta = 90^\circ$), quantum tunneling is minimized and $\rho_{xy} = \rho_{yx} = 0$. In Figs. 2a-2b (for Reviewer 3), we present the field dependences of the resistivity tensor components ρ_{zz} , ρ_{xx} , ρ_{yy} , and ρ_{zx} for $\theta = 90^\circ$, from which σ_{zz} can be calculated using Eq. (8) (see Fig. 2c for Reviewer 3). Because ρ_{zz} is one or two orders of magnitude greater than all other tensor elements ρ_{xx} , ρ_{yy} , and ρ_{zx} , the calculated $\sigma_{zz}(B)$ almost overlaps with $1/\rho_{zz}(B)$ (Fig. 2c for Reviewer 3), indicating our assumption for Eq. (10) is valid.

Fig. 2 (for Reviewer 3) (a) Field dependence of $\rho_{zz}(B)$ for $\theta = 90^\circ$. (b) Field dependence of $\rho_{xx}(B)$, $\rho_{yy}(B)$, and $\rho_{zx}(B)$ for $\theta = 90^\circ$. (c) Field dependence of $\rho_{zz}(B)$ and $1/\sigma_{zz}(B)$ for $\theta = 90^\circ$ ($B//y$). $\sigma_{zz}(B)$ is obtained through tensor

conversion using Eq. 8. (d) Fitting $\sigma_{zz}(B)$ to $\sigma_0/(1+k_1B^2)$ and $\sigma_0/(1+k_2B)$ in the low (0-2.5T) and high (3.5-9T) field regions respectively.

We have also followed Reviewer 3's suggestion and verified that $\sigma_{zz}(B)$ calculated using Eq. (8) does follow the $\sigma_0/(1+k_1B_{xy}^2)$ dependence in the low field range (0-2.5T) and the $\sigma_0/(1+k_2B_{xy})$ dependence in the high field range (3.5-9T) for Dirac band transport of $\theta = 90^\circ$, as shown in Fig. 2d (for Reviewer 3). We have added these additional analyses to the Supplementary Note 2.

For Eq. (5) in the previous manuscript, we assumed that $\sigma_{zx} = 1/\sigma_{xz}$. In general, such an assumption is not appropriate and tensor conversion (Eq. 8) should be used for the analysis, as indicated by Reviewer 3. We apologize for overlooking this and appreciate that Reviewer 3 pointed out this problem. As indicated above, we have followed Referee 3's suggestion and performed additional measurements (Fig. 1a-1f for Reviewer 3) to create resistivity tensor and re-analyzed our data as shown below.

Based on the double transport channel model discussed in the manuscript, the total Hall conductivity in YbMnBi₂ can be expressed as

$$\sigma_{zx}^{total} = w_1 \cdot \sigma_{zx}^{LL0} + w_2 \cdot \sigma_{zx}^C \quad (11)$$

where σ_{zx}^{LL0} and σ_{zx}^C represent the Hall conductivities contributed by Weyl and Dirac bands, respectively. w_1 and w_2 represent the weights of the contribution for each type of band. The Hall conductivity of the momentum relaxation channel of the Dirac bands, σ_{zx}^C , can be found from the Boltzmann transport theory, i.e.

$$\sigma_{zx}^C = \sigma_0 \frac{\omega_c \tau}{1 + (\omega_c \tau)^2} \quad (12),$$

where $\sigma_0 = \frac{ne^2\tau}{m^*}$ and $\omega_c = \frac{eB \sin \theta}{m^*}$.

For the zeroth LL's tunneling channel of the Weyl bands, its Hall conductivity σ_{zx}^{LL0} under a field oriented at an angle of θ (inset to Fig. 1c for Reviewer 3) can be expressed as the following equation according to the theoretical study by Osada [1],

$$\sigma_{zx}^{LL0}(B, \theta) = a \frac{B_y}{B_z^2} \exp\left(-b \frac{B_y^2}{B_z}\right) \quad (13),$$

where a and b are material dependent constants, $B_y = B \sin \theta$ and $B_z = B \cos \theta$.

The total Hall conductivity σ_{zx}^{total} can be obtained via taking the inverse of the resistivity tensor.

Using resistivity tensor elements ρ_{xx} , ρ_{yy} , ρ_{xy} , ρ_{zx} , and ρ_{zz} shown in Fig. 1e-1f (for Reviewer 3), we calculated the angular dependence of the total Hall conductivity $\sigma_{zx}^{total}(\theta)$ using Eq. (9) [see Fig. 1g for Reviewer 3]. $\sigma_{zx}^{total}(\theta)$ displays a $\sin\theta$ -like dependence at lower fields, but strongly deviates from it at high fields and a local minimum at $\theta = 90^\circ$ is observed when the field is above 1 T. Such an unusual evolution of $\sigma_{zx}^{total}(\theta)$ with magnetic field cannot be described by the classical transport model (Eq. 12), but can be understood by considering the zeroth LL quantum tunneling (Eq. 13): At low fields, when the zeroth and 1st LLs are not well separated, the quantum tunneling makes minor contribution so that the total Hall conductivity is dominated by the classical momentum relaxation transport of Dirac bands,

Thus it exhibits a $\sin\theta$ dependence as predicted by Eq. (12) which can be approximated to $\sigma_{zx}^C \propto \omega\tau \propto B \sin\theta$ for low fields. However, at high fields, the interlayer zeroth LL tunneling becomes important, thus leading to unusual angular dependence of σ_{zx}^{total} . The local minimum of σ_{zx}^{total} at $\theta=90^\circ$ is caused by the suppression of the 2D Landau level quantization when the field is oriented close to the in-plane direction ($\theta=90^\circ$). This interpretation is verified by the quantitative fit of σ_{zx}^{total} to Eq. 11, as shown in Fig. 1g (for Reviewer 3).

Additionally, the newly measured $\rho_{xy}(\theta)/\rho_{yx}(\theta)$ data also exhibit anomalies attributable to the zeroth LL transport. As seen in Fig. 1c and 1d (for Reviewer 3), $\rho_{xy}(\theta)$ and $\rho_{yx}(\theta)$ exhibit a $\cos\theta$ -like dependence in the low field range (<6T), but strongly deviates from it and displays sharp kinks near 70° and 110° when the field is increased above 6T. It is worth noting that the sharp increases of $\rho_{zx}(\theta)$ and $\rho_{zz}(\theta)$ appears in the angle range of 70° - 110° (see Fig. 1e and 1f), indicating the anomalous behaviors of $\rho_{zx}(\theta)$, $\rho_{zz}(\theta)$ and $\rho_{xy}(\theta)$ all have the same origin, *i.e.* the zeroth LL's contribution to transport. However, the quantitative fit of the $\rho_{xy}(\theta)$ data is difficult due to the multiple-band complexity.

Again, we thank Reviewer #3 for pointing out this important issue. We have added the above discussions and new data to the revised manuscript.

After revising the above points, I would recommend the publication of this manuscript.

We have addressed all the points raised by Referee 3 and revised the manuscript accordingly. We hope Referee 3 will be satisfied with our revision. Finally, we want to thank Referee 3 again for taking so much time to review our manuscript and giving us important suggestions to improve our manuscript.

Reference

- [1] T. Osada, Anomalous Interlayer Hall Effect in Multilayer Massless Dirac Fermion System at the Quantum Limit, *J. Phys. Soc. Jpn.* **80**, 033708 (2011)
- [2] M. Sato, K. Miura, S. Endo, S. Sugawara, N. Tajima, K. Murata, Y. Nishio and K. Kajita, Transport Phenomenon of Multilayer Zero-Gap Conductor in the Quantum Limit, *J. Phys. Soc. Jpn.* **80**, 023706 (2011)
- [3] H. K. Pal and D. L. Maslov, Necessary and sufficient condition for longitudinal magnetoresistance, *Phys. Rev. B* **81**, 214438 (2010)
- [4] L. Li, K. Wang, D. Graf, L. Wang, A. Wang and C. Petrovic, Electron-hole asymmetry, Dirac fermions, and quantum magnetoresistance in BaMnBi₂, *Phys. Rev. B* **93**, 115141 (2016)
- [5] J. Liu, J. Hu, H. Cao, Y. Zhu, A. Chuang, D. Graf, D. J. Adams, S. M. A. Radmanesh, L. Spinu, I. Chiorescu, et al., Nearly Massless Dirac fermions hosted by Sb square net in BaMnSb₂, *Sci. Rep.* **6**, 30525 (2016)
- [6] J.Y. Liu, J. Hu, D. Graf, S. M. A. Radmanesh, D. J. Adams, Y. L. Zhu, G. F. Chen, X. Liu, J. Wei, I. Chiorescu, et al., Discovery of a topological semimetal phase coexisting with ferromagnetic behavior in Sr_{1-y}Mn_{1-z}Sb₂ (y~0.08), *arXiv:1507.07978* (2015)
- [7] E. H. Sondheimer, The Boltzman Equation for Anisotropic Metals, *Proceedings of the Royal Society of London. Series A. Mathematical and Physical Sciences* **268**, 100-108 (1962)
- [8] D. Stroud and F. P. Pan, Effect of isolated inhomogeneities on the galvanomagnetic properties of solids, *Phys. Rev. B* **13**, 1434-1438 (1976)

REVIEWERS' COMMENTS:

Reviewer #2 (Remarks to the Author):

In the revised manuscript and reply, the authors have addressed my concern properly. I am satisfied with the improvement and recommend the manuscript for publication in Nature Communications.

Reviewer #3 (Remarks to the Author):

The authors have significantly revised the manuscript by showing some additional transport data as well as correcting the expressions of conductivity. I think the present manuscript is now technically sound.

Response to Reviewer #2's comments:

In the revised manuscript and reply, the authors have addressed my concern properly. I am satisfied with the improvement and recommend the manuscript for publication in Nature Communications.

We thank Reviewer 2 for taking the time to review our revised manuscript and for his/her recommendation for publication of our manuscript in Nature Communications.

Response to Reviewer #3's comments:

The authors have significantly revised the manuscript by showing some additional transport data as well as correcting the expressions of conductivity. I think the present manuscript is now technically sound.

We thank Reviewer 3 for taking the time to review our revised manuscript and for his/her recommendation for publication of our manuscript in Nature Communications.